# Wavenumber-Resolved Spectral Gating for Diffusion Models of Two-Dimensional Turbulence

## Abstract

We study unconditional generative modeling of two-dimensional Kolmogorov flow with denoising diffusion. A plain U-Net diffusion model reproduces the point-wise statistics of turbulent vorticity fields but distorts the inertial-range spectrum and the spectral fluxes that carry energy and enstrophy across scales. We add a wavenumber-resolved spectral gate: a small Fourier-domain bottleneck that learns a per-channel multiplicative correction over a coarse radial-wavenumber grid, conditioned on the diffusion noise level and a continuous log-viscosity parameter. The gate is paired with physics losses on enstrophy, modal spectral amplitude, vorticity structure functions, integral length, and spectral flux. On forced 2D Kolmogorov flow at seven viscosity regimes spanning a factor of nine in Reynolds number, and against four unconditional baselines (a plain U-Net, a squeeze-and-excite bottleneck, an FNO-block bottleneck at $2.8\times$ the parameters, and a standalone Fourier neural operator), the gated model (WRSG, 2.51M parameters) reduces the aggregate log-spectral distance to direct numerical simulation (DNS) by 59% relative to the plain U-Net, beats the larger FNO block on that metric, and raises inverse-energy and forward-enstrophy cascade recovery from 10% and 21% to 49% and 89%. Adding an FNO branch to the gate (WRSG+FNO, 7.27M) is best overall, reaching 84% and 110% cascade recovery and the lowest flux error. A factorial ablation separates the gate, which routes spectral power and corrects the large scales and cascade transport, from the losses, which correct small-scale point statistics, and shows a strong positive interaction: the gate alone barely lowers the spectral distance and the losses alone raise it, but together they cut it by more than half. A one-parameter scalar variance recalibration, fit on a regime-stratified calibration split, brings the ensemble to the 90% coverage target, raising empirical coverage from 0.72–0.85 to 0.899–0.902. The single viscosity-conditioned model generalizes to unseen and out-of-range Reynolds numbers, and the gate, though not the vorticity-tuned loss weights, transfers to a different forcing geometry and to a passive scalar, evidence that the architectural component is general rather than turbulence-specific. We report two negative results: the physics losses worsen the large-scale integral length relative to the gate alone, and no variant fully recovers the DNS cascade peaks. Code is available at `https://tinyurl.com/3jcjp46z`.

## 1 Introduction

Two-dimensional turbulence is a standard testbed for studying the inverse energy cascade and the forward enstrophy cascade (Kraichnan, 1967; Batchelor, 1969). It is also a demanding testbed for generative models of physical fields. The data is high-dimensional, multi-scale, and constrained by conservation laws whose violation is easy to detect: a generator that fits the mean and variance but distorts the spectrum is exposed by a single log–log plot. A generative model of 2D turbulence can therefore be held to a physical, rather than perceptual, standard.

Denoising diffusion produces high-quality unconditional generators for natural images (Ho et al., 2020; Song et al., 2021; Karras et al., 2022), and recent work applies it to turbulent flows (Shu et al., 2023; Lippe

et al., 2023; Kohl et al., 2024). Most of that work studies *conditional* surrogates that predict the next state from the previous one. We study the *unconditional* setting, in which the model draws independent samples from the steady-state distribution given only the flow's viscosity. This setting is harder, because it has no previous-state input to anchor the prediction—consistent with Suresh Babu et al. (2025), who find conditional diffusion recovers fine-scale spectral-tail statistics of quasi-geostrophic turbulence better than guided-unconditional generation—and it is the relevant one for ensemble forecasting and data assimilation (Rozet & Louppe, 2023), which need independent draws from the flow's invariant measure rather than a single trajectory, so that the cost of an ensemble becomes the cost of sampling instead of many forward integrations. Our contribution is not the setting but a mechanism for it.

In the unconditional setting, the dominant failure mode of a U-Net diffusion model is not amplitude or variance, which are usually correct after enough training. It is the *spectrum*: the distribution of power across wavenumbers, and the spectral fluxes that move energy and enstrophy between scales. We document this failure and address it with two ingredients, evaluated against four unconditional baselines (Section 4.3).

First, we introduce a Fourier-domain bottleneck, the *wavenumber-resolved spectral gate*, which amplifies or attenuates the spectral bands the rest of the network mishandles. Second, we add a set of soft physics losses on enstrophy, modal spectral amplitude, vorticity structure functions, integral length, and spectral flux. A single model conditioned on a continuous log-viscosity scalar covers all seven Reynolds regimes.

Our contributions are:

1. A wavenumber-resolved spectral gate that, with matched physics losses, improves the spectral and cascade fidelity of an unconditional diffusion generator without changing the U-Net's spatial structure. One continuous-viscosity model spans seven regimes.

2. A factorial ablation that separates the gate (spectral routing, large scales, and cascade transport) from the losses (small-scale point statistics), revealing a positive interaction in which the two components are unhelpful or harmful in isolation yet together more than halve the log-spectral distance.

3. A parameter-efficiency comparison in which the gated model (WRSG, 2.51M) beats an FNO-block bottleneck (7.08M) and a standalone FNO operator (4.97M) on the spectrum, the structure functions, and the forward-enstrophy cascade at 35% of the FNO block's parameter count.

4. A hybrid extension (WRSG+FNO) that adds an FNO branch to the gate and is best overall, together with a one-parameter scalar variance recalibration of the per-pixel ensemble interval and an analysis of where every variant still fails.

5. Generality evidence beyond the training configuration: the single conditioned model interpolates and extrapolates to unseen Reynolds numbers, and the gate transfers to a different forcing geometry and to a passive scalar. A mechanism analysis, a root-cause analysis, and hyperparameter sweeps (bins, rank, loss weights, $\sigma_{\max}$, sampling steps) show the gate is the general component while the loss weights are field-specific.

## 2 Background

**Denoising diffusion.** Given a clean field $x_0$, the forward process adds Gaussian noise at scale $\sigma$ to produce $x_\sigma = x_0 + \sigma\,\epsilon$ with $\epsilon \sim \mathcal{N}(0, I)$. A network learns to denoise $x_\sigma$, and sampling starts from pure noise and anneals $\sigma$ to zero. We use the EDM parameterization of Karras et al. (2022), in which the network output is

$$D(x;\sigma) = c_{\text{skip}}(\sigma)\,x + c_{\text{out}}(\sigma)\,f_\theta(c_{\text{in}}(\sigma)\,x; c_{\text{noise}}(\sigma))\,, \tag{1}$$

where $f_\theta$ is the trainable network and the $c_{(\cdot)}(\sigma)$ are fixed preconditioning coefficients that keep the network input and target at unit scale across noise levels. Training minimizes a weighted mean-squared error between $D(x_\sigma;\sigma)$ and the clean field $x_0$, so the network predicts $x_0$ directly rather than the noise $\epsilon$. Predicting $x_0$ lets us apply physics losses to a field with a meaningful spectrum at every training step (Section 4.5). For sampling we use DPM-Solver++ 2M (Lu et al., 2022), a deterministic high-order solver for the reverse process.

**Cascades in 2D turbulence.** Forced 2D turbulence has two conserved quantities in the inviscid limit, energy and enstrophy (mean-square vorticity). Kraichnan (1967) showed that this produces a dual cascade: energy moves toward large scales (the inverse energy cascade, at wavenumbers $k < k_f$ below the forcing scale $k_f$), while enstrophy moves toward small scales (the forward enstrophy cascade, at $k > k_f$). The *spectral flux* $\Pi(k)$ measures the net rate at which a conserved quantity crosses wavenumber $k$; its sign and peak location encode the cascade directions. A generative model can match the spectrum $E(k)$ at each $k$ yet still misrepresent the fluxes, because fluxes depend on nonlinear phase relationships between modes, not on per-mode power alone. We therefore evaluate both per-mode statistics and the fluxes (Section 4.6).

# 3 Related Work

**Generative models for fluid flows.** Shu et al. (2023) apply a physics-informed diffusion model to flow-field reconstruction from sparse measurements. Lippe et al. (2023) introduce PDE-Refiner, which uses iterative denoising to stabilize next-step PDE solvers. Kohl et al. (2024) benchmark autoregressive conditional diffusion against deterministic baselines for turbulent flow. These works study conditional generation given a previous state; we study the unconditional setting.

**Unconditional turbulence generation.** Closest to our setting, Whittaker et al. (2024) train an unconditional diffusion model of two-dimensional turbulence and verify Kolmogorov scaling through structure functions, energy spectra, and velocity distributions. Their target is the inverse-energy-cascade regime: a flow forced at $k_f \sim 40$ with generation on a $256^2$ grid, an $E(k) \sim k^{-5/3}$ spectrum, inverse-cascade structure-function exponents ($\zeta_2 = 2/3$, $\zeta_3 = 1$), and velocity-increment agreement reported as $D_{\mathrm{KL}} = 0.033$ and $0.007$ at separations $r = 2, 4$. We instead force at $k_f = 4$ and target the forward enstrophy cascade and the spectral *fluxes* on a $128^2$ grid, scored by log-spectral distance, flux RMSE, and cascade recovery (Section 4.6): the two occupy different cascade regimes with no shared metric, so the numbers are complementary rather than a head-to-head, and reaching their $k^{-5/3}$ inverse range is easier than reaching the steep enstrophy-cascade tails (vorticity-spectrum slopes $-2.64$ to $-5.62$) our conditioned model must span. Sambamurthy & Chattopadhyay (2025) target the same inertial-range spectral degradation, but through a power-law ("lazy") noise schedule rather than a Fourier-domain correction, in autoregressive emulation. Our contribution is distinct from both: the wavenumber-resolved gate, its superadditive interaction with the physics losses, and the capacity-matched factorial that isolates the two; we release the eight-metric suite of Section 4.6 to enable a matched comparison.

**Neural operators.** Fourier neural operators (Li et al., 2021; Kovachki et al., 2023) apply learned spectral convolutions over a fixed set of modes and are a standard architectural baseline for grid-based PDE problems. We use them in two roles: as a single FNO block in the U-Net bottleneck, and as a standalone denoiser backbone. The proposed gate differs from an FNO block in three ways: it is multiplicative rather than transform-and-project, it is conditioned on the diffusion noise level and viscosity, and its parameters sit in the conditioning MLP rather than in per-mode learnable weights. This last point makes it more parameter-efficient than the FNO block. We also combine the two, since they act on different parts of the problem (Section 4.3).

**Physics-informed losses.** Soft physics losses originate with physics-informed neural networks (Raissi et al., 2019), which penalize the residual of a governing PDE at collocation points, and recur in hybrid and learned solvers that embed conservation, divergence, or spectral constraints to keep long rollouts stable (Kochkov et al., 2021; Stachenfeld et al., 2022). Most such penalties act in physical space on a PDE residual or a point-wise field, and target a trajectory the model must integrate. Ours instead act on distributional diagnostics of the generated ensemble—the per-scale statistics and cascade fluxes of Section 4.5—which is what an unconditional generator must match, and, in our factorial, they help only in combination with the spectral gate rather than on their own.

**Calibrated uncertainty.** Our uncertainty estimate is the per-pixel ensemble percentile band, rescaled by a single scalar fit on a held-out split to match a target marginal coverage. We call this a scalar variance recalibration rather than a conformal method: unlike conformal prediction (Vovk et al., 2005; Angelopoulos

& Bates, 2021), which builds prediction sets from a nonconformity score at a chosen quantile with a finite-sample coverage guarantee, our rescaling fits one variance scale and makes no such guarantee. Post-hoc recalibration of this kind is standard for probabilistic forecasts, where marginal coverage and proper scores such as the continuous ranked probability score (Matheson & Winkler, 1976; Gneiting & Raftery, 2007) are the usual reported quantities; we report calibration error on a disjoint test split and, because a single global scale cannot correct a wavenumber-dependent under-dispersion, also resolve calibration by scale (Section 5).

## 4 Setup

### 4.1 Direct numerical simulation

We solve the 2D incompressible Navier–Stokes equations in vorticity form on a doubly periodic domain $[0, 2\pi]^2$ with Kolmogorov forcing,

$$\partial_t \omega + (\mathbf{u} \cdot \nabla)\omega = \nu \nabla^2 \omega - \alpha \omega + f, \qquad f = k_f^2 \cos(k_f y), \qquad (2)$$

where $\omega = \nabla \times \mathbf{u}$ is the scalar vorticity, $\mathbf{u}$ is recovered from $\omega$ through the streamfunction, $\nu$ is the kinematic viscosity, $\alpha = 0.1$ is a linear (Ekman) drag that removes energy at large scales, and $k_f = 4$ is the forcing wavenumber. We integrate on a $128 \times 128$ grid with a pseudo-spectral RK4 scheme at timestep $\Delta t = 10^{-3}$. The convective product is evaluated with full 3/2-rule zero-padding (Orszag, 1971) to remove aliasing of high-wavenumber product modes into the resolved range.

We simulate seven Reynolds regimes at viscosities $\nu \in \{0.005, 0.007, 0.010, 0.013, 0.018, 0.024, 0.032\}$, chosen to span a factor of nine in Reynolds number and so test a viscosity-conditioned generator across regimes. After a 4000-step spinup we save 1000 decorrelated snapshots per regime, one every 200 steps (7000 snapshots in total). From these snapshots, the integral length $L$ (from the radial spectrum) and the 2D Taylor microscale $\lambda = \sqrt{2\langle|\mathbf{u}|^2\rangle/\langle\omega^2\rangle}$ give integral Reynolds numbers $\text{Re}_{\text{int}} = u_{\text{rms}}L/\nu$ from 593 down to 63, and Taylor-microscale Reynolds numbers $\text{Re}_\lambda = u_{\text{rms}}\lambda/\nu$ from 318 down to 33 (Figure 1).

Figure 1 shows the time-averaged vorticity power spectra $|\hat{\omega}(k)|^2$ (the enstrophy spectrum, $k^2$ times the kinetic-energy spectrum). The fitted slopes over the inertial window $k \in [6, 14]$ run from $-2.64$ at the least viscous regime to $-5.62$ at the most viscous, all with $R^2 \geq 0.979$, steepening with viscosity as the dissipation scale moves into the resolved range. These are steeper than the Kraichnan enstrophy-cascade prediction (Kraichnan, 1967; Boffetta & Ecke, 2012) (a $k^{-1}$ vorticity spectrum, equivalently a $k^{-3}$ kinetic-energy spectrum), reflecting the short inertial range and modest Reynolds number at $N = 128$ rather than an asymptotic cascade.

### 4.2 Data splits

We hold out 15% of all snapshots (1050 fields) as a fixed test set, with the split fixed by a seed-0 permutation before any training. The same test set evaluates every variant at every training seed. The remaining 5950 snapshots form the training pool, split per seed into inner-train (90%) and inner-val (10%); the inner-val is used only for training-time monitoring, never for reported metrics. Fixing the test split before training, and keeping it independent of the per-seed split, removes the train–test contamination that arises when a per-run training split also sets the per-run evaluation split. The physics metrics of Section 4.6 are computed per seed by comparing 256 unconditionally generated fields against 256 held-out test fields, with the generated set's regime composition matched to that of the test set, so that no training data enters the evaluation.

### 4.3 Architecture and ablations

The denoiser $f_\theta$ in equation 1 maps a noisy single-channel $128^2$ vorticity field to a denoised one through a three-level U-Net (Ronneberger et al., 2015) with group normalization. A $3 \times 3$ convolution lifts the input to 48 channels; the encoder then applies a residual block at each level while halving the resolution and doubling the width, giving feature maps of $48 \times 128^2$, $96 \times 64^2$, and $192 \times 32^2$, and a symmetric skip-connected decoder inverts this back to a single output channel. (A width of 48 base channels was the smallest at which the plain U-Net reached stable validation loss in pilot runs.) Every residual block is conditioned by a 128-dimensional

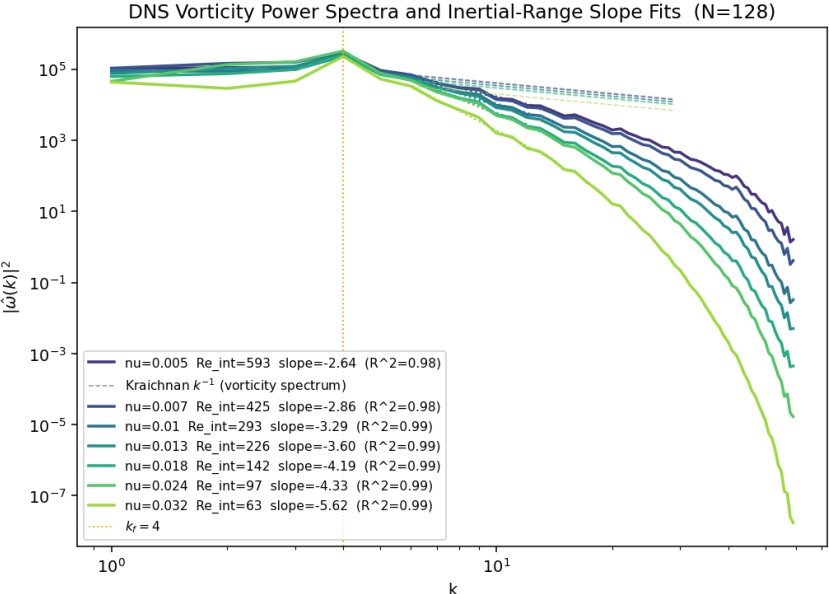

Figure 1: Time-averaged DNS vorticity power spectra $|\hat{\omega}(k)|^2$ at $N = 128$ ($k^2$ times the kinetic-energy spectrum), one curve per viscosity regime. The inertial-range fit window is $k \in [6, 14]$. The dashed line marks the Kraichnan enstrophy-cascade reference for the vorticity spectrum, $|\hat{\omega}(k)|^2 \sim k^{-1}$ (equivalently a $k^{-3}$ kinetic-energy spectrum). Legend Reynolds numbers are integral-scale, from the time-averaged snapshots.

embedding that fuses sinusoidal embeddings of the noise level $c_{\text{noise}}(\sigma)$ and of a continuous viscosity scalar, the $z$-scored $\log \nu$, so one model spans all seven regimes on a shared physical scale rather than through a discrete regime index. The bottleneck, acting on the 192-channel $32 \times 32$ feature map, is the only component that differs across the U-Net variants. We study a $2 \times 2$ factorial design (gate present or absent $\times$ losses present or absent) plus extra baselines and one extension:

- `vanilla`: U-Net, no bottleneck, no physics loss.

- `se`: squeeze-and-excite bottleneck (Hu et al., 2018), no physics loss. Tests whether any conditional bottleneck helps.

- `fno`: a single FNO block (Li et al., 2021) with 8 modes per axis as a bottleneck, no physics loss. Tests whether a spectral architecture helps inside the U-Net.

- `fno_operator`: a standalone Fourier neural operator denoiser (12 modes, width 64, four spectral layers), trained with the same EDM objective and no physics loss. An external architecture baseline: an FNO used as the denoiser backbone rather than as a U-Net bottleneck.

- `wrsd_gate`: the gate, no physics loss. Isolates the gate (WRSG-gate).

- `wrsd_phys`: no bottleneck, with physics losses. Isolates the losses (WRSG-phys).

- `wrsd`: gate plus losses. The main model (wavenumber-resolved spectral gate, WRSG).

- `wrsd_fno`: gate plus an FNO branch plus losses (WRSG+FNO). The hybrid extension.

The factorial design holds capacity fixed across the comparisons that matter: `vanilla` and `wrsd_phys` share an architecture (2.32M parameters), and `wrsd` and `wrsd_gate` share an architecture (2.51M parameters). Comparing `vanilla` with `wrsd_phys` (or `wrsd_gate` with `wrsd`) therefore isolates the losses, and comparing `vanilla` with `wrsd_gate` (or `wrsd_phys` with `wrsd`) isolates the gate.

### 4.4 The wavenumber-resolved gate

Let $h \in \mathbb{R}^{B \times C \times H \times W}$ be the U-Net bottleneck feature map, and let $\hat{h} = \mathcal{F}(h)$ denote its 2D orthonormal Fourier transform. The gate multiplies the Fourier coefficients by a learned per-band factor,

$$\hat{h}_{\text{out}}(k_x, k_y) = \hat{h}(k_x, k_y) \odot \left(1 + s \cdot \left[2\,\sigma(g_{c,b(k)}) - 1\right]\right), \tag{3}$$

and the bottleneck adds the gated signal back to its input as a residual, returning $h + \mathcal{F}^{-1}(\hat{h}_{\text{out}})$ to the spatial domain. In the Fourier domain the bottleneck therefore applies the single per-band multiplier $2 + s \cdot \left[2\,\sigma(g_{c,b(k)}) - 1\right]$, centered at 2 rather than 1. Here $\sigma$ is the logistic sigmoid, $b(k)$ bins the radial wavenumber $k = \sqrt{k_x^2 + k_y^2}$ into 16 uniformly spaced bins, $g \in \mathbb{R}^{C \times 16}$ is the per-channel, per-bin gate-logit matrix with entries $g_{c,b}$, and $s \in \mathbb{R}$ is a learnable scalar initialized to zero. A small MLP maps the conditioning embedding to a low-rank factorization of $g$, $g_{c,b} = \sum_{r=1}^{R} a_{r,c} \beta_{r,b} + d_c$ with rank $R = 2$ and a per-channel bias $d_c$, so the conditioning MLP emits the $R(C + 16) + C$ values of this factorization rather than the $16C$ of a dense per-(channel, bin) logit map. The gate (the MLP head plus the scalar $s$) adds about 0.19M parameters, 8% over the plain U-Net.

The form of equation 3 is the minimal correction consistent with the symmetries of the statistics it targets, not a free-form spectral transform. The spectrum and the fluxes are isotropic functions of $|k|$, so the correction is indexed by the radial wavenumber rather than by $(k_x, k_y)$; this also makes the parameter count independent of grid size. The correction is multiplicative because spectral bias is a multiplicative distortion of the spectrum, an under- or over-prediction of power within a band, rather than an additive offset. The map $2\sigma(\cdot) - 1 \in (-1, 1)$ centers the correction at zero, and the scale $s$ is initialized to zero, so the gate's correction starts inactive and grows only as needed; the centering also lets the gate raise ($g > 0$) or lower ($g < 0$) the power in a band, redistributing the spurious high-$k$ energy we document in Section 5. Because the gate is applied as a residual, the bottleneck is not the identity at initialization but a uniform doubling of its input ($s = 0$ gives the per-band multiplier 2); this fixed factor is immaterial to the factorial comparison, since the gate's learnable correction still starts at zero, the downstream decoder is trained end-to-end, and the no-gate variants carry no bottleneck at all. Because the bin index depends only on $|k|$, the gate map keeps conjugate symmetry and the inverse transform stays real. We use 16 bins because the gate operates on the $32 \times 32$ bottleneck feature map, whose radial spectrum spans about 22 shells (the diagonal Nyquist wavenumber of a $32^2$ grid); 16 bins coarsen these enough to pool statistics within a band while still separating the inertial and dissipation ranges. Sensitivity to this choice is discussed in Section 6.

**The hybrid bottleneck.** The gate controls per-shell amplitude but does not mix Fourier modes, and inter-scale transfer (the fluxes) is a cross-mode effect. The WRSG+FNO bottleneck adds an FNO spectral convolution branch alongside the gate, returning $\text{gate}(h) + s_{\text{F}}\,\text{FNO}(h)$ with a learnable branch weight $s_{\text{F}}$ initialized to one. The FNO branch supplies the cross-mode mixing the gate lacks; Section 5 shows it closes most of the remaining cascade-flux gap, at the cost of the FNO block's parameters.

### 4.5 Diffusion schedule and losses

We standardize the data to zero mean and unit variance with a single global mean and standard deviation computed over the full dataset (two dataset-level scalars, so the test contribution is negligible), which fixes the EDM data scale at $\sigma_{\text{data}} = 1$ and calibrates the preconditioning coefficients in equation 1. We set $\sigma_{\text{max}} = 20$ rather than the EDM default of 80 because at $\sigma = 80$ the first sampling step is dominated by the raw network output, which our data poorly constrains; in pilot runs $\sigma_{\text{max}} = 80$ produced near-zero samples. The training noise distribution $\log \sigma \sim \mathcal{N}(-1.0, 1.2^2)$ is recentered from the EDM default to match this reduced range. The remaining diffusion settings ($\sigma_{\text{min}} = 0.002$, DPM-Solver++ 2M with 50 steps) follow the EDM defaults.

We train with AdamW (Loshchilov & Hutter, 2019) at learning rate $2 \times 10^{-4}$, weight decay $10^{-5}$, and batch size 32 for 160 epochs ($N_{\text{steps}} = 26{,}880$), with cosine annealing after an eight-epoch warmup. We keep an EMA of the weights with decay $d = \min\left(1 - 1/(0.2\,N_{\text{steps}}), 0.999\right)$; at our run length the 0.999 cap binds, giving a window of about $10^3$ steps ($\approx 4\%$ of training). The conventional fixed $d = 0.9999$ ($\approx 10^4$ steps) over-smoothed the weights and degraded samples in pilot runs.

For the physics-loss variants (`wrsd_phys`, `wrsd`, `wrsd_fno`) we add soft penalties on physical diagnostics of the predicted clean field $x_0$, recovered from $x_\sigma$ through the EDM preconditioning,

$$\mathcal{L}_{\mathrm{phys}} = \lambda_E \left\| \langle x_0^2 \rangle_{\mathrm{pred}} - \langle x_0^2 \rangle_{\mathrm{true}} \right\|_1 + \lambda_S \left\| |\hat{x}_0|_{\mathrm{pred}} - |\hat{x}_0|_{\mathrm{true}} \right\|_1 + \lambda_C \, \mathcal{D}_{\mathrm{struct}}$$
$$+ \mathbb{1}[\sigma < 1] \left( \lambda_L \, \mathcal{D}_{\mathrm{len}} + \lambda_\Pi \, \mathcal{D}_{\mathrm{flux}} \right), \tag{4}$$

with weights $\lambda_E = \lambda_S = \lambda_C = 0.05$, $\lambda_L = 0.10$, $\lambda_\Pi = 0.07$. The first term penalizes enstrophy-magnitude error and the second modal spectral-amplitude error; $\mathcal{D}_{\mathrm{struct}}$ is the log-space error of the order-2 and order-3 vorticity structure functions over separations $r = 1, \ldots, 8$, which targets the small-scale increments the radial gate tends to distort; $\mathcal{D}_{\mathrm{len}}$ is the integral-length error, a large-scale constraint that counteracts the small-scale pull of the other terms; and $\mathcal{D}_{\mathrm{flux}}$ is the peak-normalized energy- and enstrophy-flux error. The flux variant `wrsd_fno` and the main model `wrsd` use all five terms; `wrsd_phys` uses the first four (no flux term). The integral-length and flux terms are applied only at low noise ($\sigma < 1$), where the predicted clean field is informative enough about the large scales and the nonlinear transport for these higher-order diagnostics to give a useful gradient; at high noise the $x_0$ estimate is too smooth for them to be meaningful. We set the amplitude weights to 0.05 so that $\mathcal{L}_{\mathrm{phys}}$ is roughly one tenth of the EDM denoising loss at initialization, large enough to shape the spectrum but small enough not to override the denoising objective. These weights, the bin count, the MLP rank, and $\sigma_{\mathrm{max}}$ were fixed a priori rather than tuned on the held-out test set: the weights by this initialization-scale rule, the bin count by the radial-shell count of the bottleneck (Section 4.4), the rank and $\sigma_{\mathrm{max}}$ by pilot runs, both judged from generated samples without the test labels. The test set is used for the final reported numbers and for the sensitivity sweeps in Appendix D: the LSD is flat across the bin count and the rank (Table 15), and the a-priori loss-weight scale ($1\times$) and $\sigma_{\mathrm{max}}$ (20) are also the LSD-minimizing settings of their sweeps, with the result insensitive to them within a band around the chosen values (Tables 16, 19).

## 4.6 Evaluation metrics

We report eight physics metrics. Four are point statistics:

- **Aggregate log-spectral distance (LSD)**, $\frac{1}{|K|} \sum_k |\log_{10} \langle E_{\mathrm{pred}}(k) \rangle - \log_{10} \langle E_{\mathrm{true}}(k) \rangle|$, where $E(k)$ is the dealiased *vorticity power spectrum* $|\hat{\omega}(k)|^2$ and $\langle \cdot \rangle$ averages over the field set. This is our single summary of spectral fidelity.

- **Vorticity structure functions** of order 2 and 3, as log-RMSE over separations $r = 1, \ldots, N/3$ (the well-resolved range, wider than the $r \leq 8$ used by the loss term). We use the absolute-moment form $S_p^\omega(r) = \langle |\omega(\mathbf{x} + r\mathbf{e}) - \omega(\mathbf{x})|^p \rangle$, with $\mathbf{e}$ an axis-aligned unit vector, averaged over the two axis-aligned directions; the absolute form keeps $S_3$ positive and stable to estimate from small samples, unlike the signed velocity structure functions of Frisch (1995).

- **Integral length relative error**. The length $L = \pi \sum_k k^{-1} |\hat{\omega}(k)|^2 / \sum_k |\hat{\omega}(k)|^2$ is computed from the full (non-dealiased) vorticity power spectrum; it is therefore an enstrophy-weighted length scale, used here as a consistent relative-fidelity metric rather than the classical energy-based integral scale.

Four are spectral-transport metrics:

- **Energy flux RMSE**, normalized by $\max_k |\Pi_E^{\mathrm{DNS}}(k)|$, with $\Pi_E$ in the sign convention of Boffetta & Ecke (2012): $\Pi_E(k) < 0$ for $k < k_f$ marks the inverse energy cascade.

- **Enstrophy flux RMSE**, defined the same way for $\Pi_Z$.

- **Inverse energy cascade recovery**, the fraction of the inverse-cascade flux the model recovers, clipped to $[0, 200]\%$:

$$100 \cdot \frac{\sum_{0 < k < k_f} \max(-\Pi_E^{\mathrm{pred}}(k), 0)}{\sum_{0 < k < k_f} \max(-\Pi_E^{\mathrm{DNS}}(k), 0)}.$$

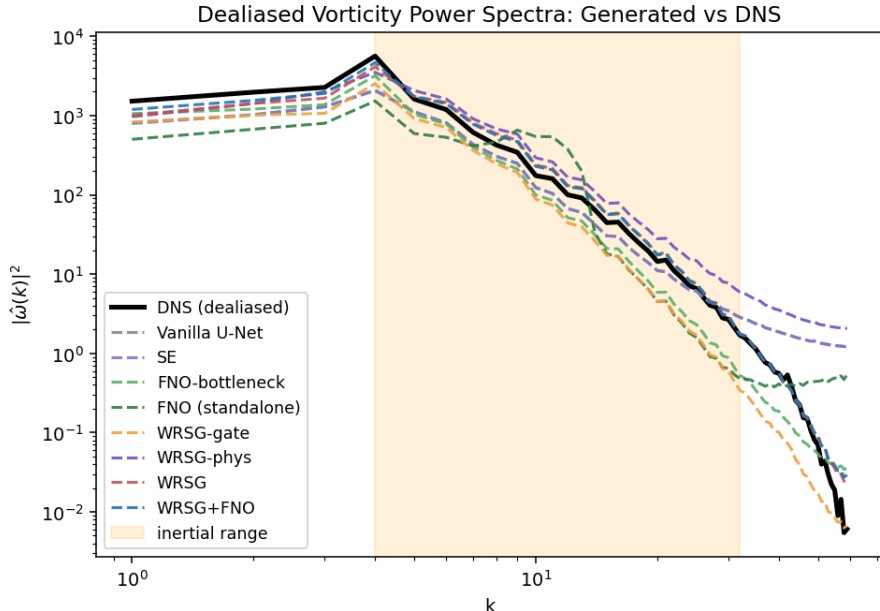

Figure 2: Dealiased vorticity power spectra $|\hat{\omega}(k)|^2$, generated vs DNS, each averaged over its field ensemble pooled across all five seeds. The shaded band marks the inertial range, from the forcing wavenumber $k_f = 4$ to $k = 32$. WRSG and WRSG+FNO track the DNS curve across the resolved range. WRSG-phys (losses without the gate) overpredicts power at the smallest scales; the standalone FNO and the plain baselines sit above DNS in the dissipation tail; WRSG-gate (gate without losses) underpredicts the high-$k$ tail.

- **Forward enstrophy cascade recovery**, defined the same way for $\Pi_Z$ at $k > k_f$. A value above $100\%$ is an overshoot of the DNS flux, not additional improvement, so we treat the flux RMSEs as the primary transport metric and the cascade-recovery percentages as their sign-and-magnitude summary.

For each metric we run five seeds $(29, 47, 89, 101, 149)$, report the seed mean with $95\%$ bootstrap confidence intervals over 2000 resamples (Efron, 1979), and run paired Wilcoxon signed-rank tests (Wilcoxon, 1945) against each baseline.

### 4.7 Uncertainty quantification

We draw a 32-member ensemble for each of 64 test conditions. For each (pixel, condition) the ensemble gives a median and a 5th–95th percentile band. We fit a single scalar $\alpha$ on a $50\%$ calibration split to match $90\%$ marginal coverage, then report empirical coverage and the continuous ranked probability score (Matheson & Winkler, 1976; Gneiting & Raftery, 2007) on the held-out $50\%$.

## 5 Results

### 5.1 Spectral fidelity

Figure 2 shows the time-averaged generated spectra against DNS. WRSG and WRSG+FNO track the DNS curve across the resolved range, including the forcing peak at $k = k_f = 4$, the inertial slope, and the dissipation tail. The other variants fail at different scales: the plain baselines and the standalone FNO place excess power in the dissipation tail, the losses without the gate (WRSG-phys) overpredict the smallest scales, and the gate without the losses (WRSG-gate) underpredicts the high-$k$ tail. Each failure has a distinct scale signature, which the ablation in Section 5.3 explains.

Table 1: Headline physics metrics, mean over five seeds. Lower is better for LSD, structure functions, length error, and flux RMSEs; closer to 100% is better for cascade recovery. Best per metric in bold, second-best underlined. Column headers: `van` (vanilla), `se`, `fno` (FNO block), `f-op` (standalone FNO), `gate` (WRSG-gate), `phys` (WRSG-phys), `WRSG`, `+FNO` (WRSG+FNO). 95% bootstrap confidence intervals are shown in Figure 3 and tabulated in Appendix A.

| Metric | van | se | fno | f-op | gate | phys | WRSG | +FNO |
|---|---|---|---|---|---|---|---|---|
| LSD aggregate ↓ | 0.744 | 0.745 | 0.553 | 0.757 | 0.688 | 0.925 | 0.304 | **0.293** |
| Vort. $S_2$ log-RMSE ↓ | 0.260 | 0.254 | 0.224 | 0.280 | 0.302 | 0.054 | 0.044 | **0.033** |
| Vort. $S_3$ log-RMSE ↓ | 0.354 | 0.348 | 0.329 | 0.417 | 0.424 | 0.131 | 0.071 | **0.061** |
| Integral length rel. err. ↓ | 0.039 | 0.037 | 0.033 | 0.228 | **0.013** | 0.150 | 0.112 | 0.073 |
| Energy flux RMSE ↓ | 0.138 | 0.137 | 0.086 | 0.139 | 0.119 | 0.120 | 0.083 | **0.038** |
| Enstrophy flux RMSE ↓ | 0.163 | 0.162 | 0.109 | 0.177 | 0.144 | 0.122 | 0.081 | **0.046** |
| Inv. energy cascade rec. % ↑ | 10.0 | 10.3 | 45.9 | 8.8 | 21.4 | 24.8 | 48.6 | **83.8** |
| Fwd. enstrophy cascade rec. % ↑ | 20.7 | 20.9 | 41.3 | 5.8 | 25.9 | 71.2 | 89.0 | **110.0** |

## 5.2 Headline metrics

Table 1 summarizes the eight metrics across all eight variants. WRSG+FNO is best on seven of the eight metrics and fifth on the remaining one (integral length); WRSG is second on those same seven. The two proposed models lead the four baselines on every metric except integral length, where the gate alone (`wrsd_gate`) is best and the physics losses move the proposed models behind the plain baselines, the trade-off we analyze in Sections 5.3 and 6.

WRSG beats the larger FNO-block bottleneck on the spectrum, both structure functions, and forward-enstrophy recovery, and ties it on energy flux at a fraction of its size; Section 5.5 gives the per-metric numbers and the parameter-efficiency comparison. The standalone FNO operator (`fno_operator`) is the weakest variant overall: an FNO used as the denoiser backbone, with no U-Net and no physics loss, does not match even the plain U-Net on the spectrum or the cascade. The spectral architecture helps as a bottleneck inside the U-Net, not as a replacement for it.

**Paired tests.** WRSG improves over `vanilla` on seven of eight metrics at paired Wilcoxon $p = 0.0625$, the floor for $n = 5$ (every seed agrees on the sign of the effect), with large effect sizes (Cohen's $|d|$ from 7.7 on energy flux to 41.5 on the $S_2$ structure function); it is worse than `vanilla` only on integral length. WRSG+FNO improves on WRSG on the flux and cascade metrics ($p = 0.0625$; energy flux $\Delta = -0.044$, $d = -7.2$) but is statistically tied on LSD ($\Delta = -0.010$, $p = 0.125$, $d = -1.1$): the FNO branch buys cascade transport, not aggregate spectral fidelity. Full per-metric differences with bootstrap CIs are in Appendix A.

## 5.3 Ablation: gate, losses, and their interaction

Reading each effect against `vanilla` (Figure 3):

- The gate alone (`wrsd_gate`) cuts energy flux RMSE by 14% and enstrophy flux RMSE by 11%, doubles inverse-energy cascade recovery (21.4% vs 10.0%), and nearly removes the integral-length error (1.3% vs 3.9%, the best of any variant). It worsens the structure functions ($S_2$ from 0.260 to 0.302). The gate acts as a spectral router: it corrects how power is distributed across scales but does not fit small-scale point statistics on its own.

- The losses alone (`wrsd_phys`) sharply lower the structure-function log-RMSE ($S_2$ 0.054 vs 0.260) and raise forward-enstrophy recovery (71.2% vs 20.7%), but they *raise* LSD to 0.925 (the worst of any variant) and worsen the integral length. Pushing the network toward small-scale and transport fidelity with no gate to route the spectral correction, the losses distort the aggregate spectrum.

The interaction is the main finding of the ablation. The gate alone lowers LSD only slightly (gate 0.688 vs vanilla 0.744), and the losses alone raise it to 0.925; adding the two individual effects would predict no

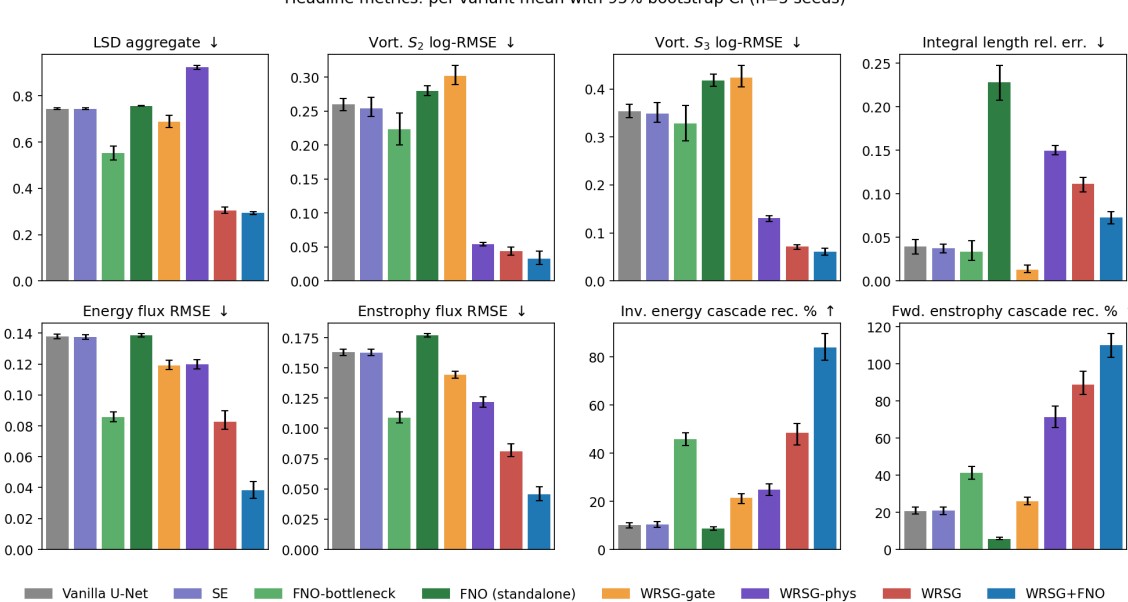

Figure 3: Per-variant mean with 95% bootstrap CI over five seeds, across the eight headline metrics. The gate (WRSG-gate) reduces integral-length error and cascade-flux error but is worst on structure functions. The losses (WRSG-phys) are strong on structure functions and forward-enstrophy recovery but raise LSD and integral-length error. The full model (WRSG) and the hybrid (WRSG+FNO) inherit the gains of both.

improvement. Yet together they reach 0.304, a 59% reduction. The gate gives the network a place to apply a clean spectral correction, and the losses tell it what correction to make; the losses are harmful without the gate to route them, and the gate does little without the losses to drive it. WRSG+FNO then adds the FNO branch and improves the flux and cascade metrics further (energy flux 0.038, inverse-energy recovery 84%), at the FNO block's parameter cost. Figure 4 shows the flux curves; the structure-function curves are in Appendix G.

### 5.4 What the gate learns, and what it fixes

Two diagnostics make the gate–loss interaction of Section 5.3 mechanistic: one reads the learned gate, the other locates the error the gate removes.

**The learned gate.** Figure 5 shows the gate's per-band multiplier $m_b(\sigma) = 1 + s\,[\,2\,\sigma(g_{c,b}) - 1\,]$ from equation 3 (the factor applied to the bottleneck spectrum, before the residual add), averaged over channels and the five seeds, as a function of the diffusion noise level $\sigma$ and the radial band. The learned scale is $s = 0.995$, so the gate is far from the identity. The correction is band-selective, varying across the sixteen radial bins rather than scaling the spectrum uniformly, and it is active across the whole noise schedule: the mean deviation $\langle |m_b - 1| \rangle$ is 0.40 for $\sigma < 1$ and 0.53 for $\sigma \geq 1$. The gate is a per-shell amplitude control the network applies at every denoising step, expressed through the low-rank $R(C + 16) + C$ factorization of its per-(channel, bin) logits.

**The error it fixes.** Figure 6 locates the spectral distortion. The right panel plots the generated-to-DNS spectrum ratio per band: the plain U-Net under-predicts the inertial range (ratio 0.5–0.7) and over-predicts the dissipation tail, where the ratio reaches about $50\times$ at the highest resolved wavenumber. WRSG stays within a factor of about 1.5 over the bulk of the resolved range and within $\approx 2.7\times$ at the dealiasing edge. The left and center panels trace this to the single-step denoiser. We add noise at a grid of $\sigma$ to held-out clean fields, denoise once, and measure the per-band log-spectral error $|\log_{10} E_{\text{pred}}(k) - \log_{10} E_{\text{true}}(k)|$. The plain denoiser's error concentrates at high $\sigma$ and high $k$ (mean high-$k$ error 0.575 at $\sigma > 1$, against 0.323

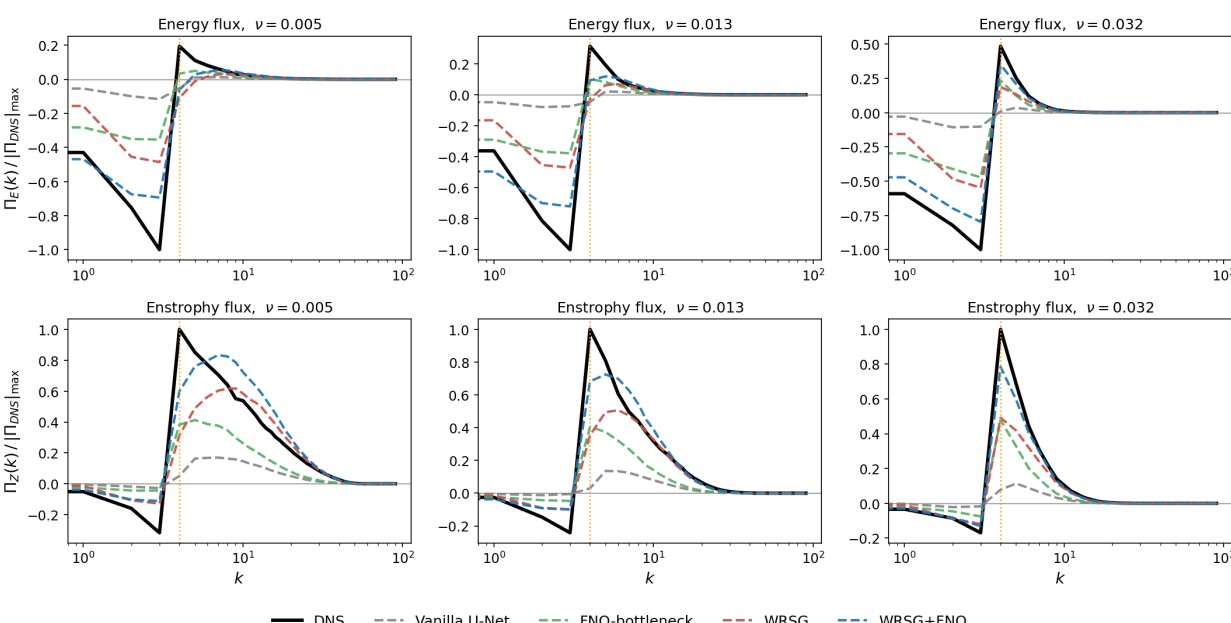

Figure 4: Normalized spectral fluxes for three representative regimes (least, mid, and most viscous). Top row: energy flux $\Pi_E(k)$. Bottom row: enstrophy flux $\Pi_Z(k)$. Curves are normalized by the peak DNS flux magnitude in each panel; the plot shows DNS and the four variants that carry the flux story, pooled across all five seeds (the ablation variants are in Figure 3 and the tables). WRSG+FNO recovers the largest fraction of the DNS flux; no variant fully matches the peaks. The residual gap is a limitation of current diffusion-based generators (Section 6).

for WRSG; the two are equal, $\approx 0.03$, at low $\sigma$). The bias enters at the high-noise steps that fix the global structure, where the convolutional denoiser over-estimates the dissipation-tail amplitude, and it survives the reverse process into the end-to-end excess.

This identifies the cause. It is not the diffusion objective: the per-step denoising MSE is nearly identical across all U-Net variants (Appendix F), yet their spectra differ, so the error is frequency-localized in a way the MSE does not register. It is not that any spectral representation helps: the standalone, fully spectral FNO operator is the weakest variant on five of the eight metrics (the fluxes, the cascade, and the integral length; Table 1) and does not match even the plain U-Net on the spectrum or the cascade. It is the convolutional U-Net's high-wavenumber amplitude bias at high noise. This is a spectral bias in the sense of a frequency-localized systematic error, but opposite in direction to the low-frequency *preference* Rahaman et al. (2019) document for coordinate-based networks: rather than under-representing high frequencies, our denoiser *over*-predicts the dissipation-tail amplitude, because at high noise the small-scale content of the clean field is weakly constrained by the noised input and the network fills that band with excess power. The two share only the notion of a frequency-localized error; the sign is reversed. The gate, a multiplicative correction in exactly that band, is what removes it.

**Where the gate acts: the forward map, not the gradient.** A natural hypothesis is that the gate helps by absorbing or damping the physics-loss gradient. We test it directly: for the gated (`wrsd`) and ungated (`wrsd_phys`) models we backpropagate the physics loss at a grid of noise levels and measure the gradient reaching the deepest encoder feature, and the share carried by the gate's own parameters (Figure 7). The hypothesis fails. The physics-loss gradient grows monotonically with $\sigma$ and is nearly identical with and without the gate (the gate's own parameters carry under 0.3% of the total gradient energy (squared norm)

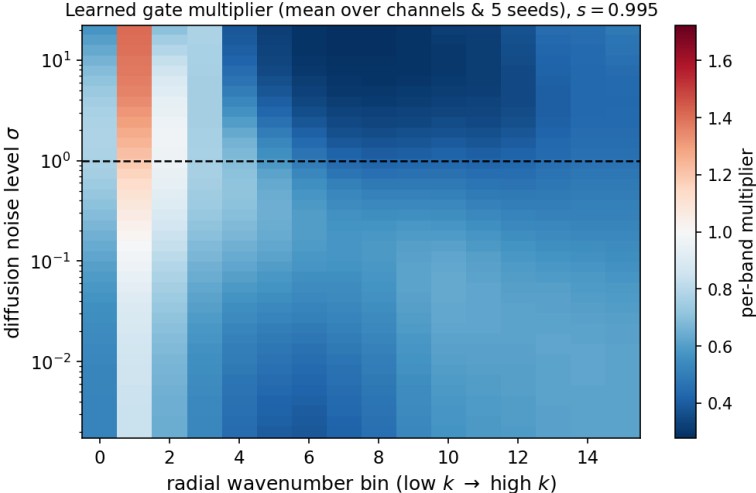

Figure 5: Learned gate multiplier $m_b(\sigma)$ (mean over channels and five seeds), per radial band (horizontal) and diffusion noise level (vertical, log scale), for WRSG at the mid viscosity. The dashed line marks the $\sigma < 1$ threshold below which the integral-length and flux losses are active. The correction is band-selective and active at every noise level, not a single global scale.

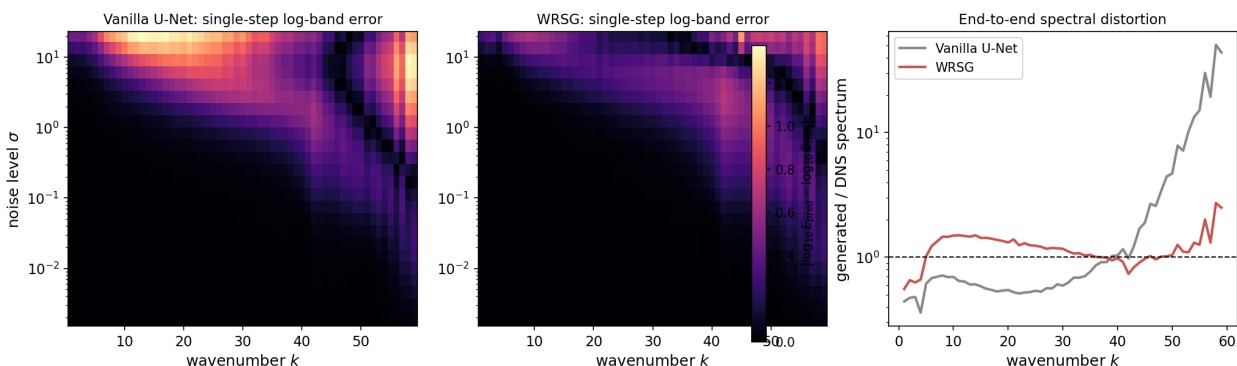

Figure 6: Origin of the spectral distortion. Left, center: single-step denoising error in log-spectral space, $|\log_{10} E_{\text{pred}} - \log_{10} E_{\text{true}}|$, against wavenumber $k$ and noise level $\sigma$, for the plain U-Net and WRSG at $\nu = 0.005$. The plain denoiser's error concentrates at high $\sigma$ and high $k$; the gate suppresses it. Right: the end-to-end generated/DNS spectrum ratio. The plain U-Net's dissipation tail greatly over-predicts DNS at the highest resolved wavenumbers, where WRSG instead stays close to DNS across the resolved range (ratios in Section 5.4).

and slightly *raise* the backbone gradient rather than damping it). The gate's effect is therefore in the *forward* computation, not in the gradient: it supplies a per-shell multiplicative degree of freedom that makes the spectral correction the losses demand expressible at all, which a purely spatial decoder cannot realize without trading away the aggregate spectrum. This is why the physics losses corrupt the spectrum on their own (`wrsd_phys`, LSD 0.925) and why neither component helps alone (Section 5.3).

## 5.5 Parameter efficiency

The FNO-block baseline uses 7.08M parameters; WRSG uses 2.51M, or 35% of it. At that size WRSG beats the FNO block on LSD (0.304 vs 0.553; $\Delta = -0.249$, $p = 0.0625$), on both structure functions, and on

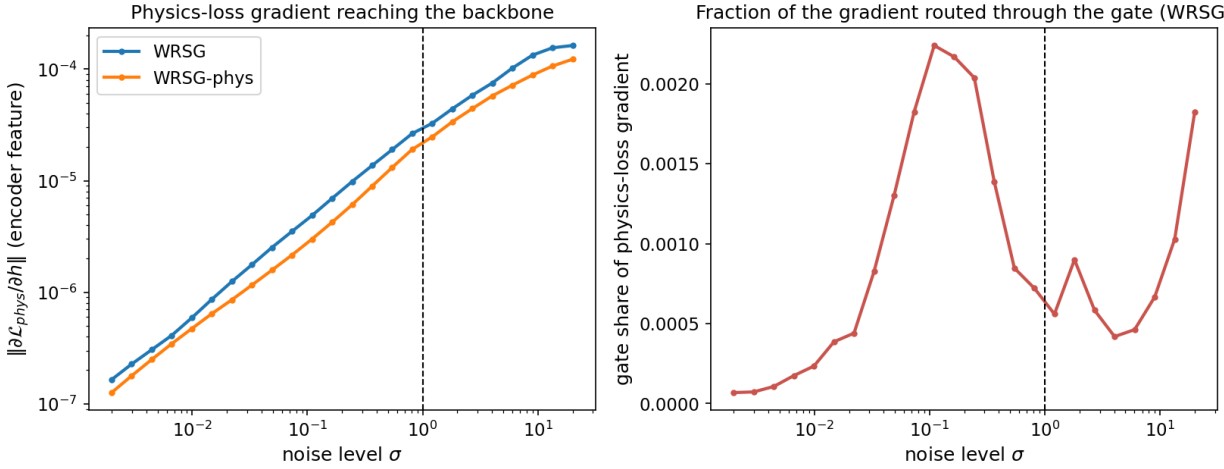

Figure 7: Physics-loss gradient flow across noise levels (mean over five seeds). Left: the norm of the physics-loss gradient reaching the deepest encoder feature, for the gated (WRSG) and ungated (WRSG-phys) models; it grows with $\sigma$ and the gate does not reduce it. Right: the fraction of the physics-loss gradient energy (squared norm) carried by the gate's own parameters in WRSG, which stays under 0.3%. The gate's role is the forward per-shell correction (Figure 5), not a change to how the loss gradient flows.

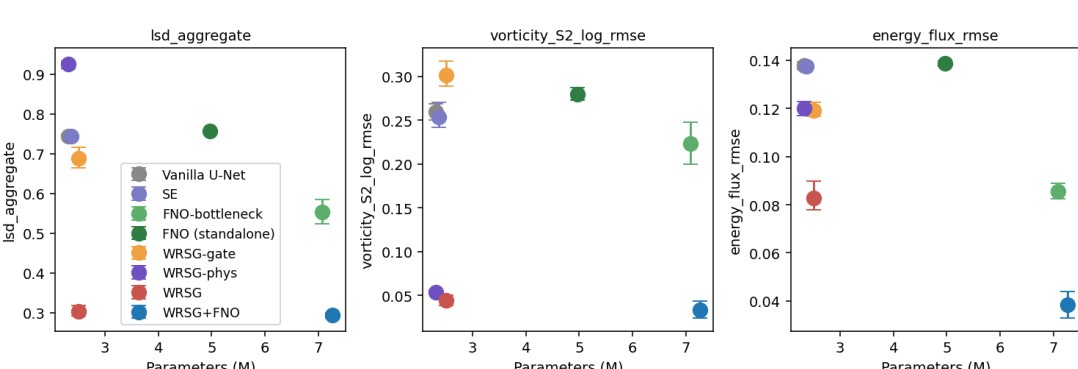

Figure 8: Parameter–performance Pareto fronts for three metrics, with 95% bootstrap CIs. WRSG (2.51M) is on the front for LSD and the structure functions, beating the 2.8×-larger FNO block. WRSG+FNO (7.27M) extends the front on energy flux. The standalone FNO operator is off the front on every metric.

forward-enstrophy recovery, and matches it on energy flux RMSE (0.083 vs 0.086; $\Delta = -0.003$, $p = 0.625$, not significant). WRSG+FNO (7.27M) is the strongest model and uses about the same parameter count as the FNO block, so the gain over the FNO block on the flux and cascade metrics comes from the gate-plus-loss combination, not from added capacity. Figure 8 shows these fronts.

## 5.6 Per-regime breakdown

Table 2 shows the LSD improvement over `vanilla` holds across all seven regimes and grows in absolute terms with viscosity.[1] At the most viscous regime ($\nu = 0.032$, $\mathrm{Re}_{\mathrm{int}} = 63$) WRSG cuts LSD from 3.41 to 1.80 (a 47% reduction); at the least viscous ($\nu = 0.005$) it goes from 0.56 to 0.32 (43%). `vanilla` struggles

---

[1]The per-regime LSD here pools snapshots within a single regime, whereas the headline LSD in Table 1 pools all seven regimes into one ensemble-mean spectrum before taking the log-spectral distance. Because the LSD is a nonlinear function of

Table 2: Per-regime LSD aggregate. The improvement over `vanilla` holds across all seven regimes and grows in absolute terms with viscosity, where the steep dissipation tail makes spectral fitting harder for every method. WRSG and WRSG+FNO are first or second in every regime.

| Variant | $\nu$=.005 | .007 | .010 | .013 | .018 | .024 | .032 |
|---|---|---|---|---|---|---|---|
| vanilla | 0.555 | 0.701 | 0.994 | 1.343 | 1.890 | 2.287 | 3.408 |
| se | 0.557 | 0.704 | 0.995 | 1.339 | 1.893 | 2.297 | 3.415 |
| fno | 0.528 | 0.578 | 0.634 | 0.795 | 1.004 | 1.292 | 1.990 |
| fno_operator | 0.727 | 0.775 | 0.852 | 1.028 | 1.258 | 1.492 | 2.282 |
| wrsd_gate | 0.677 | 0.709 | 0.724 | 0.823 | 0.896 | 1.180 | 1.874 |
| wrsd_phys | 0.701 | 0.837 | 1.159 | 1.511 | 2.076 | 2.426 | 3.552 |
| **wrsd** | 0.317 | 0.313 | 0.386 | **0.534** | **0.791** | **1.036** | **1.801** |
| **wrsd_fno** | **0.308** | **0.306** | **0.375** | 0.549 | 0.803 | 1.045 | 1.823 |

most at high $\nu$, where the dissipation tail steepens, and the gate compensates by moving capacity away from the low-amplitude high-$k$ modes. WRSG and WRSG+FNO hold first and second place in every regime, with the largest absolute margin over `vanilla` at the most viscous regimes; the FNO block and the gate-only model also keep LSD below 2 throughout but never reach the top two. These seven regimes are the training viscosities, so Table 1 and Table 2 measure in-distribution fidelity; the next section tests the same continuous-viscosity model at viscosities held out from training, both between the training grid points and outside the training range.

### 5.7 Out-of-distribution viscosity

Because the model is conditioned on a continuous log-viscosity rather than a regime index, it can be queried at viscosities absent from training. We test this on fresh DNS at five interior (interpolation) viscosities between training grid points, $\nu \in \{0.0085, 0.0115, 0.0155, 0.0210, 0.0275\}$, and two outside the training range $[0.005, 0.032]$ (extrapolation), $\nu \in \{0.0045, 0.040\}$. For each, the conditioning is the $z$-scored $\log \nu$ under the training grid's statistics (not refit); we sample 192 fields per seed and score against the new DNS. Table 3 reports the aggregate LSD. WRSG improves on the plain U-Net at every unseen viscosity. At the interpolation points its LSD tracks the in-distribution trend of Table 2 (for instance 0.338 at $\nu = 0.0085$, between the training values at $\nu = 0.007$ and $\nu = 0.010$), so the continuous conditioning interpolates smoothly rather than only memorizing the seven trained regimes. Extrapolation is asymmetric: just below the training range ($\nu = 0.0045$) WRSG reaches 0.275, comparable to its best in-distribution regime, whereas above the range ($\nu = 0.040$, the most viscous and steepest-tailed regime) all models degrade and WRSG reaches 1.975 – still well below the plain U-Net's 3.554, but the largest error of the sweep, consistent with the in-distribution pattern that high viscosity is the hard end (Table 2).

---

the ensemble-mean spectrum and the regimes differ by orders of magnitude in spectral power, the headline value is not the average of the per-regime values and can fall below all of them.

Table 3: Out-of-distribution viscosity: aggregate LSD (mean over five seeds) at viscosities absent from training, ordered by $\nu$. Interpolation points lie between training grid points; extrapolation points lie below (0.0045) and above (0.040) the training range $[0.005, 0.032]$. Lower is better; best per row in bold. WRSG improves on the plain U-Net at every unseen viscosity.

| $\nu$ (kind) | vanilla | fno | WRSG | +FNO |
|---|---|---|---|---|
| 0.0045 (extrap) | 0.535 | 0.562 | 0.275 | **0.262** |
| 0.0085 (interp) | 0.824 | 0.624 | **0.338** | 0.339 |
| 0.0115 (interp) | 1.191 | 0.673 | **0.459** | 0.466 |
| 0.0155 (interp) | 1.553 | 0.849 | **0.580** | 0.593 |
| 0.0210 (interp) | 2.001 | 1.104 | **0.831** | 0.844 |
| 0.0275 (interp) | 2.552 | 1.415 | **1.168** | 1.179 |
| 0.0400 (extrap) | 3.554 | 2.153 | **1.975** | 1.988 |

## 5.8 Transfer to a different flow and a different field

Two transfers test whether the gain is specific to the Kolmogorov vorticity setup. Both keep the gate and the losses unchanged from the main model; only the flow or the field changes.

**A different flow: cellular forcing.** We replace the 1D Kolmogorov shear forcing with a 2D cellular forcing $f = k_f^2(\cos k_f x + \cos k_f y)$ at $k_f = 6$, which drives a different mean structure at a forcing scale the gate was not tuned for, and retrain the gate/loss factorial over three viscosity regimes. The pattern of the main flow reproduces (Table 4): the gate alone lowers the LSD (0.316 against the plain U-Net's 0.382) and lifts inverse-energy cascade recovery to 51% (the plain U-Net reaches 40%), and adding the physics losses lifts it to 98% and overshoots the forward cascade to 146%, a larger overshoot than even the hybrid produces on the main flow (110%), at a small LSD cost. The gate and the losses carry over to a different forcing configuration without retuning.

Table 4: Cellular-forcing flow ($k_f = 6$), mean over three seeds. As on the main flow, the gate lowers the LSD and the physics losses recover most of the inverse-energy cascade; here the full model overshoots the forward cascade. Lower is better for LSD and flux; closer to 100% is better for cascade recovery.

| Variant | LSD | Energy flux | Inv. cascade % | Fwd. cascade % |
|---|---|---|---|---|
| vanilla | 0.382 | 0.108 | 39.7 | 51.8 |
| wrsd_gate | **0.316** | 0.088 | 51.1 | 57.0 |
| wrsd | 0.403 | **0.048** | **97.8** | 146.4 |

**A different field: passive scalar.** We transfer the gate to a passive scalar $\theta$ advected by the Kolmogorov flow with an imposed mean gradient (Warhaft, 2000), $\partial_t \theta + (\mathbf{u} \cdot \nabla)\theta = \kappa \nabla^2 \theta - vG$ (where $G$ is the imposed mean scalar gradient and $v$ the component of the velocity $\mathbf{u}$ along it), a different field obeying a different (advection–diffusion) PDE, over four scalar-diffusivity regimes with continuous $\log \kappa$ conditioning (the velocity-specific flux terms are dropped). Here the two components separate (Table 5): the gate alone transfers and lowers the LSD (0.529 against 0.713), but the physics losses, with weights calibrated for vorticity, sharpen the structure functions ($S_2$ 0.127 against 0.302) while raising the aggregate LSD to 0.866, so the full set does not transfer to the new field as configured. The architectural component, the gate, is general; the soft-loss weights are field-specific. A loss sweep on the scalar localizes the cause and points to the fix (Appendix D, Table 18): the vorticity-specific structure, enstrophy, and integral-length terms drive the scalar regression, while restricting the loss to its field-agnostic spectral term alone matches the gate-only transfer. The transfer recipe for a new field is the gate plus the spectral loss.

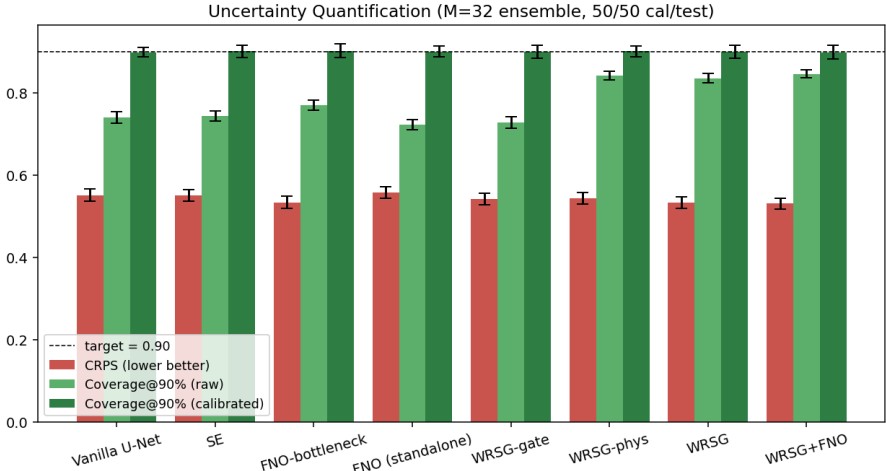

Figure 9: Uncertainty quantification. Raw and variance-rescaled per-pixel 90% coverage and CRPS across variants, with the calibration split stratified across regimes. The single-scalar rescaling reaches the 0.90 target; CRPS is lowest for WRSG+FNO.

Table 5: Passive-scalar transfer (mean over three seeds). The gate alone lowers the LSD; the vorticity-tuned losses sharpen the structure functions but raise the LSD, so they do not transfer to the new field without reweighting. The $S_2, S_3$ here are scalar-increment structure functions. Lower is better.

| Variant | LSD | $S_2$ log-RMSE | $S_3$ log-RMSE | Integral len. err. |
|---|---|---|---|---|
| vanilla | 0.713 | 0.302 | 0.499 | 0.238 |
| wrsd_gate | **0.529** | 0.333 | 0.547 | **0.194** |
| wrsd | 0.866 | **0.127** | **0.203** | 0.363 |

**A different resolution.** We retrain the gate/loss factorial on a $64^2$ grid (bottleneck $16^2$) over three viscosity regimes. The gate bins the normalized radial wavenumber, so its form is unchanged; following the rule of Section 6 we hold the bin count at the same fraction of the bottleneck's radial extent, halving it from 16 to 8 as the bottleneck shrinks from $32^2$ to $16^2$. The main-flow pattern reproduces at the new resolution (Table 6): the gate alone gives the lowest LSD (0.521 against the plain U-Net's 0.629), and adding the physics losses recovers the inverse-energy cascade (98.8% against 38.9%) and cuts the energy-flux error more than fourfold (0.029 against 0.134), with the same forward-cascade overshoot (110%) and small LSD cost seen on the $128^2$ flow. The gain is not tied to a single grid.

Table 6: Resolution transfer to a $64^2$ grid (mean over three seeds), gate bins scaled $16 \rightarrow 8$ with the bottleneck. The gate lowers the LSD and the physics losses recover the inverse-energy cascade, reproducing the $128^2$ behavior. Lower is better for LSD and flux; closer to 100% is better for cascade recovery.

| Variant | LSD | Energy flux | Inv. cascade % | Fwd. cascade % |
|---|---|---|---|---|
| vanilla | 0.629 | 0.134 | 38.9 | 46.8 |
| wrsd_gate | **0.521** | 0.102 | 52.9 | 51.7 |
| wrsd | 0.539 | **0.029** | **98.8** | 110.2 |

## 5.9 Calibrated uncertainty

The raw per-pixel 5–95 band gives 0.72–0.85 empirical coverage (per-variant means), below the 90% target (Figure 9). The one-parameter variance rescaling, a single scalar fit on a calibration split stratified across the seven regimes, raises coverage to 0.899–0.902 on the disjoint test split: every variant reaches the target (WRSG 0.900, WRSG+FNO 0.899, the FNO block 0.902). Stratifying the split matters because the under-dispersion and the scale it needs both vary with viscosity; on a regime-matched split one global scalar suffices for the per-pixel marginal coverage. CRPS clusters in $[0.531, 0.558]$, lowest for WRSG+FNO and WRSG (0.531, 0.533); a deep ensemble that pools the five seeds lowers it to 0.52 for both proposed models, since averaging over independently trained scores sharpens the predictive distribution.

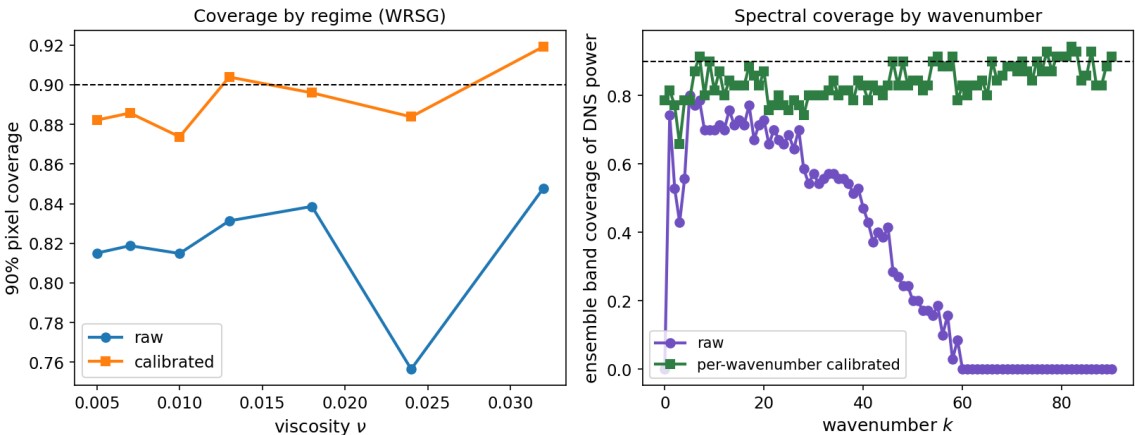

Figure 10: Uncertainty resolved by regime and scale (WRSG). Left: raw and calibrated 90% pixel coverage per viscosity regime; the calibrated coverage varies across regimes. Right: raw and per-wavenumber-calibrated coverage of the DNS spectral power by wavenumber. The raw band is well calibrated at low $k$ but under-dispersed at the small scales; a separate per-shell calibration restores coverage across the spectrum.

**Where the residual gap lives, and closing it.** Resolving the WRSG calibration by viscosity regime and by wavenumber (Figure 10, Table 7) locates the under-dispersion that a single $\alpha$ cannot absorb. The per-regime calibrated coverage ranges 0.874–0.919 and the fitted scale $\alpha$ from 1.20 to 1.45, so one global $\alpha$ is a compromise across regimes; the CRPS broadly falls from 0.663 at $\nu = 0.005$ to 0.410 at $\nu = 0.032$, since the more viscous, lower-Reynolds fields are less variable. By wavenumber, the raw 5–95 band covers the DNS band power at $\approx 0.70$ for the resolved shells $1 \leq k \leq 20$ but only $\approx 0.23$ for $k > 20$: the under-dispersion concentrates at the small scales. Letting the calibration vary with scale closes this gap. A per-wavenumber rescaling, a separate $\alpha_k$ fit per radial shell on the calibration split, raises the small-scale ($k > 20$) coverage from 0.23 to 0.84 on the small per-regime calibration set (about ten fields per regime). That 0.84 is calibration-sample-limited rather than a hard floor. On a larger pooled calibration set ($n = 128$ fields, $n_{uq} = 256$) the same multiplicative per-shell rescaling already reaches 0.89, and a split-conformal band—which adds an absolute width fit on the calibration split and so is not defeated by the near-zero small-scale ensemble spread that a multiplicative scale cannot widen—reaches 0.92 (both averaged over WRSG and WRSG+FNO, on the disjoint test split). The residual gap is thus a matter of calibration resolution and calibration-sample size, not irreducibly missing spread.

Table 7: Per-regime uncertainty for WRSG: calibrated 90% pixel coverage, the fitted scale $\alpha$, and CRPS. Coverage and $\alpha$ both vary with viscosity, so a single global $\alpha$ is a compromise.

| $\nu$ | .005 | .007 | .010 | .013 | .018 | .024 | .032 |
|---|---|---|---|---|---|---|---|
| Calibrated coverage | 0.882 | 0.886 | 0.874 | 0.904 | 0.896 | 0.884 | 0.919 |
| Scale $\alpha$ | 1.25 | 1.25 | 1.20 | 1.30 | 1.25 | 1.45 | 1.30 |
| CRPS | 0.663 | 0.637 | 0.596 | 0.558 | 0.505 | 0.589 | 0.410 |

Qualitative samples (Appendix E) and training curves (Appendix F) are in the appendix: all U-Net variants produce realistic vorticity fields, with the trade-offs already present in the metrics.

## 6 Limitations

**The losses worsen the integral length.** The physics losses sharpen the small scales and the cascade but degrade the large-scale integral length: WRSG-gate reaches an integral-length error of 0.013, while adding the losses raises it to 0.112 for WRSG and 0.073 for WRSG+FNO, both worse than the plain U-Net (0.039). The integral-length term in equation 4 reduces this regression but does not remove it. We trace the cause to the structure-function loss and map the trade-off in Appendix D (Table 17): the integral length trades directly against the small-scale and transport fidelity that term provides, a controllable Pareto trade-off rather than a tuning artifact. A practitioner who needs the large-scale length above all should drop the structure-function loss or use the gate without the losses.

**Residual cascade gap.** No variant fully matches the DNS flux peaks (Figure 4). WRSG+FNO recovers 84% of the inverse energy cascade flux but *overshoots* the forward enstrophy cascade to 110%; WRSG reaches 49% and 89%. Matching the fluxes, which depend on nonlinear transport, with a model trained only on snapshots and with no hard conservation constraint likely needs an explicit conservation loss or a hybrid generator–solver; a short pass of the true solver does correct the cascade (Appendix J), but at a structure-function cost, so it repairs the transport rather than removing the trade-off. We tempered the forward overshoot with a smaller FNO-branch weight in pilot runs, but that gave back the inverse-cascade recovery the branch provides, so we report the neutral setting and the overshoot as a trade-off. Up-weighting the flux loss does not recover the cascade peaks, so they are not reachable by a larger soft penalty and would need a hard constraint.

**Residual small-scale spread.** A single global variance scale, calibrated on a regime-stratified split, reaches the 90% per-pixel coverage target (Section 5, "Calibrated uncertainty"). The residual imperfection is spectral rather than in the pixel marginal: the raw ensemble is under-dispersed at the small scales. A per-wavenumber rescaling on the small per-regime calibration set reaches 0.84 there; with a larger pooled calibration set the same rescaling reaches 0.89 and a split-conformal band reaches 0.92 (Section 5), so the small-scale coverage is effectively at target once the calibration is given enough held-out data, and the earlier 0.84 was calibration-sample-limited rather than a hard floor. What a scalar rescaling still cannot do is make the small-scale band *sharp* while covering; a genuinely scale-resolved ensemble, not just a wider band, would be needed for that.

**Bin count and grid.** The gate bins the *normalized* radial wavenumber (Section 4.4), so its parameter count does not depend on the grid; the rule for a different resolution is to hold the bin count at a fixed fraction of the bottleneck's radial extent rather than re-tune from scratch, which Table 6 confirms on a $64^2$ grid (bin count halved to 8). The bin count, the MLP rank, the physics-loss weights, and the maximum noise level are examined in Appendix D; the sampling step count is not load-bearing, with the LSD flat above $\approx 20$ solver steps (Table 14).

**Sample size and scope.** Five seeds give a Wilcoxon floor of $p = 0.0625$ (Section 4.6); the large Cohen's $d$ values indicate the floor is reached because every seed agrees on direction, not because the per-seed effect

is marginal. All runs are at $128 \times 128$ in 2D, where the DNS is exact and the spectral diagnostics are well defined; generalization beyond this configuration is taken up in Sections 5.7 and 5.8 and in the Discussion.

# 7 Discussion

The factorial decomposition (Section 5.3) refines the usual "physics losses help" message of the PINN literature (Raissi et al., 2019): here neither the additive losses nor the multiplicative gate improves the aggregate spectrum on its own, and the gain is their interaction. The FNO branch adds a third, separable effect: cross-mode mixing that the per-shell gate cannot provide, closing most of the cascade-flux gap.

**Generality and scope.** Two design choices keep the gate from being a turbulence-specific trick. First, what helps is the particular frequency-domain correction, not a bottleneck as such: a squeeze-and-excite bottleneck of the same conditioning form does not improve the spectrum (LSD 0.745 against vanilla's 0.744), and a generic spectral block, the FNO bottleneck, helps only partway (0.553), while the multiplicative, radial, conditioned gate reaches 0.304 (Table 1). Second, the gate is defined on the radial wavenumber $|k|$ with a grid-independent parameter count (Section 4.4), carrying no 2D- or vorticity-specific structure. We test this three ways (Sections 5.7 and 5.8): the single viscosity-conditioned model generalizes to unseen and out-of-range Reynolds numbers; the gate and its losses carry over to a different flow, a 2D cellular forcing at a different forcing scale, where the inverse-cascade recovery reaches 98% and the main-flow pattern reproduces; and the gate, but not the vorticity-tuned loss weights, transfers to a passive scalar, a different field obeying a different PDE. The split on the scalar is informative: the architectural component is general, while the soft-loss weights are field-specific. Three-dimensional flows, anisotropic forcing, and experimental measurements remain untested and are the natural next targets.

We compare against the four unconditional baselines of Section 4.3 at comparable or larger capacity, the like-for-like comparison for the unconditional task. A distributional comparison against conditional next-step solvers such as PDE-Refiner (Lippe et al., 2023) or autoregressive conditional diffusion (Kohl et al., 2024) is not like-for-like: they solve a different problem, predicting the next state from the current one, so a conditional model reaches near-zero point-wise error on short horizons, while its *steady-state* distribution depends on the rollout length and can spectrally collapse over long horizons—the very failure Sambamurthy & Chattopadhyay (2025) set out to fix—and Suresh Babu et al. (2025) find the conditional advantage is on reconstruction rather than on free-running statistics. We run this comparison (Appendix I): a standard conditional emulator trained on short-time-correlated pairs at $\nu = 0.013$ reaches near-zero one-step error but drifts over a long free-running rollout to an aggregate log-spectral distance of 1.39 against DNS, against 0.49 for our unconditional samples at the same regime. The gate does not depend on that choice: it acts inside the bottleneck and drops into a conditional denoiser unchanged, so the mechanism we isolate here transfers directly to the conditional setting.

# 8 Conclusion

We presented a wavenumber-resolved spectral gate for diffusion-based unconditional generation of 2D turbulence, paired with five soft physics losses, and a hybrid extension that adds an FNO branch. Across seven Reynolds regimes with a single viscosity-conditioned model, the gated model improves over a plain U-Net on the spectrum and the cascade and is parameter-efficient against an FNO block of nearly three times the size; the hybrid is best overall. A factorial ablation shows the gate and the losses correct distinct errors and interact, and the scalar variance recalibration closes the per-pixel coverage gap. The residual cascade gap and the integral-length trade-off (Section 6) remain the main open problems. The gate is a self-contained bottleneck that can be added to an existing U-Net, and the suite of physics metrics gives a reference point for subsequent work on generative turbulence modeling.

**Broader Impact Statement**

This work studies generative modeling of an idealized fluid system, forced 2D Kolmogorov flow, at modest resolution. The setting is a benchmark rather than an application, and we see no direct dual-use concern.

The motivating use is fast generation of independent draws from a flow's steady-state (invariant) distribution. Such draws are the input to ensemble-based uncertainty quantification, to the prior in data assimilation (Appendix C gives a sparse-reconstruction example), and to the initial-condition ensembles used in geophysical and atmospheric forecasting, where the cost of a conventional ensemble is set by the number of independent forward integrations. A generator that returns a decorrelated field in tens of milliseconds, against the $\approx 1.4\,\mathrm{s}$ a pseudo-spectral solver needs to advance one decorrelation time (Appendix B), changes the cost of assembling a large ensemble, but only if its per-scale statistics and its uncertainty are trustworthy, which is what the spectral, cascade, and calibration metrics in this paper are built to measure. This per-field advantage is amortized: the $\approx 100$ GPU-hours of training must be recovered across many generated fields before the surrogate is net-cheaper than the solver, so it applies to repeated sampling from a trained model rather than to a single field. The same per-scale fidelity matters for any downstream quantity that integrates against the spectrum, such as eddy diffusivities or scale-dependent transport coefficients in subgrid models.

Two cautions bound these prospects. First, all of our evidence is on synthetic DNS of an idealized flow; deployment on a real geophysical or laboratory flow requires retraining and revalidation against the target distribution, which is anisotropic, multi-physics, and often three-dimensional, and the calibration we report would have to be re-established there. Second, a generative surrogate carries no hard conservation guarantee: the residual cascade-flux error we document (Section 6) means the samples are best treated as a statistical prior to be corrected by data or by a solver, not as a conservative simulation. We see the method as a component inside a validated assimilation or ensemble pipeline rather than a stand-alone replacement for a physical solver. Concretely, a surrogate with the imperfect flux recovery and the residual small-scale calibration we report should not drive operational forecasting or engineering decisions without fresh validation on the target system.

### Reproducibility Statement

Code, training configurations, and seed values are available at `https://tinyurl.com/3jcjp46z`, and the same code is attached as a supplementary `.zip` archive with this submission so that review does not depend on the external link resolving. Sampling at evaluation time is seeded, so the reported tables reproduce exactly from the released checkpoints. The full sweep (eight variants, five seeds, seven regimes, 160 epochs at $128^2$) takes about 100 GPU-hours. Per-variant parameter counts and training times are in Appendix B (Table 11). The DNS solver, data splits (Section 4), architecture and hyperparameters (Sections 4.3–4.5), and evaluation protocol (Section 4.6) are specified in full.

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

# A    Detailed Statistical Tests

For each headline metric we run a paired test of a proposed model against each baseline at matched seeds, reporting the mean difference with bootstrap CI, the Wilcoxon signed-rank $p$-value, and Cohen's $d$ (Cohen, 1988) on the seed-wise differences. The per-variant bootstrap intervals of Tables 8 and 9 are our primary evidence of separation; because the five seeds share a fixed, seeded evaluation protocol, the seed-wise variance is small and can inflate Cohen's $d$, so we read $d$ as a reproducibility diagnostic rather than a primary effect size. With $n = 5$ seeds the Wilcoxon floor is $p = 0.0625$, reached whenever all five seeds agree on the sign. All differences, intervals, and effect sizes are computed from the unrounded per-seed metrics, so a difference recomputed from the rounded means in Table 1 may differ in the last digit. We tabulate the main model (WRSG) against each baseline on three representative metrics in Table 10; the per-seed values for all eight metrics and both proposed models are released with the code.

## A.1    Per-variant confidence intervals

Tables 8 and 9 give all eight headline metrics for all eight variants as the seed mean with its 95% bootstrap confidence interval (2000 resamples over the five seeds), the per-variant counterpart to the headline means in Table 1. The intervals support the headline ranking: WRSG and WRSG+FNO separate from all four baselines with non-overlapping intervals on the LSD and the forward-enstrophy cascade, and on the inverse-energy cascade with the single exception of WRSG against the FNO block ($[43.2, 52.3]$ against $[43.3, 48.4]$, which overlap). WRSG and WRSG+FNO themselves overlap on LSD ($[0.292, 0.318]$ against $[0.286, 0.300]$) and separate on the flux and cascade metrics, the WRSG-versus-WRSG+FNO pattern that the paired tests below quantify.

Table 8: Headline metrics, group A (spectrum and structure): seed mean with 95% bootstrap CI over five seeds. Lower is better.

| Variant | LSD | $S_2$ | $S_3$ | Int. len. |
|---|---|---|---|---|
| vanilla | 0.744 [0.741, 0.750] | 0.260 [0.250, 0.269] | 0.354 [0.340, 0.368] | 0.039 [0.030, 0.049] |
| se | 0.745 [0.741, 0.749] | 0.254 [0.242, 0.271] | 0.348 [0.331, 0.373] | 0.037 [0.032, 0.042] |
| fno | 0.553 [0.522, 0.584] | 0.224 [0.200, 0.247] | 0.329 [0.292, 0.366] | 0.033 [0.023, 0.046] |
| fno_operator | 0.757 [0.755, 0.758] | 0.280 [0.273, 0.287] | 0.417 [0.406, 0.431] | 0.228 [0.207, 0.247] |
| wrsd_gate | 0.688 [0.663, 0.716] | 0.302 [0.289, 0.317] | 0.424 [0.404, 0.450] | 0.013 [0.009, 0.018] |
| wrsd_phys | 0.925 [0.915, 0.933] | 0.054 [0.051, 0.056] | 0.131 [0.124, 0.136] | 0.150 [0.144, 0.156] |
| wrsd | 0.304 [0.292, 0.318] | 0.044 [0.038, 0.050] | 0.071 [0.067, 0.076] | 0.112 [0.102, 0.119] |
| wrsd_fno | 0.293 [0.286, 0.300] | 0.033 [0.024, 0.043] | 0.061 [0.054, 0.069] | 0.073 [0.065, 0.080] |

Table 9: Headline metrics, group B (flux and cascade): seed mean with 95% bootstrap CI over five seeds. Lower is better for the flux RMSEs; closer to 100% is better for cascade recovery.

| Variant | E flux | Z flux | Inv. % | Fwd. % |
|---|---|---|---|---|
| vanilla | 0.138 [0.136, 0.139] | 0.163 [0.160, 0.165] | 10.0 [8.9, 11.2] | 20.7 [18.9, 22.9] |
| se | 0.137 [0.136, 0.139] | 0.162 [0.160, 0.165] | 10.3 [9.1, 11.5] | 20.9 [18.7, 22.9] |
| fno | 0.086 [0.082, 0.089] | 0.109 [0.105, 0.114] | 45.9 [43.3, 48.4] | 41.3 [37.9, 44.8] |
| fno_operator | 0.139 [0.138, 0.140] | 0.177 [0.175, 0.178] | 8.8 [8.1, 9.4] | 5.8 [5.2, 6.4] |
| wrsd_gate | 0.119 [0.117, 0.122] | 0.144 [0.141, 0.147] | 21.4 [19.1, 23.4] | 25.9 [23.8, 28.0] |
| wrsd_phys | 0.120 [0.117, 0.123] | 0.122 [0.117, 0.126] | 24.8 [22.4, 27.2] | 71.2 [65.7, 78.2] |
| wrsd | 0.083 [0.078, 0.090] | 0.081 [0.077, 0.088] | 48.6 [43.2, 52.3] | 89.0 [83.6, 96.1] |
| wrsd_fno | 0.038 [0.033, 0.044] | 0.046 [0.040, 0.051] | 83.8 [78.6, 89.7] | 110.0 [103.6, 116.3] |

Table 10: Paired tests of WRSG against each baseline on three representative metrics: the seed-wise mean difference with 95% bootstrap CI, the Wilcoxon signed-rank $p$, and Cohen's $d$, computed from the unrounded per-seed metrics. For LSD and energy flux a negative difference favors WRSG; for forward-enstrophy recovery a positive difference favors WRSG. WRSG ties the FNO block on energy flux ($p = 0.625$) and separates from every baseline on the other two.

| Baseline | $\Delta$ mean | 95% CI | Wilcoxon $p$ | Cohen's $d$ |
|---|---|---|---|---|
| *LSD aggregate* | | | | |
| vanilla | $-0.441$ | $[-0.455, -0.426]$ | 0.0625 | $-21.9$ |
| fno | $-0.249$ | $[-0.291, -0.210]$ | 0.0625 | $-4.7$ |
| fno_operator | $-0.453$ | $[-0.465, -0.439]$ | 0.0625 | $-26.6$ |
| wrsd_gate | $-0.385$ | $[-0.423, -0.348]$ | 0.0625 | $-7.9$ |
| wrsd_phys | $-0.621$ | $[-0.634, -0.610]$ | 0.0625 | $-40.9$ |
| *Energy flux RMSE* | | | | |
| vanilla | $-0.055$ | $[-0.060, -0.049]$ | 0.0625 | $-7.7$ |
| fno | $-0.003$ | $[-0.008, +0.003]$ | 0.6250 | $-0.4$ |
| fno_operator | $-0.056$ | $[-0.061, -0.050]$ | 0.0625 | $-7.7$ |
| wrsd_gate | $-0.036$ | $[-0.040, -0.032]$ | 0.0625 | $-7.4$ |
| wrsd_phys | $-0.037$ | $[-0.042, -0.032]$ | 0.0625 | $-5.6$ |
| *Forward enstrophy cascade recovery* | | | | |
| vanilla | $+68.3$ | $[+64.2, +73.9]$ | 0.0625 | $+10.9$ |
| fno | $+47.6$ | $[+44.7, +51.9]$ | 0.0625 | $+9.8$ |
| fno_operator | $+83.2$ | $[+78.3, +89.8]$ | 0.0625 | $+10.7$ |
| wrsd_gate | $+63.0$ | $[+59.3, +68.6]$ | 0.0625 | $+10.3$ |
| wrsd_phys | $+17.7$ | $[+15.1, +20.3]$ | 0.0625 | $+5.3$ |

The hybrid extension improves on WRSG on the transport metrics (energy flux $\Delta = -0.044$, $[-0.049, -0.040]$, $p = 0.0625$, $d = -7.2$; forward enstrophy recovery $\Delta = +21.0$, $p = 0.0625$, $d = +4.6$) but is tied with WRSG on LSD ($\Delta = -0.010$, $p = 0.125$, $d = -1.1$).

# B  Compute Profile

Table 11: Per-variant parameter count and wall-clock training time. Training time and peak memory are means over five seeds, on one V100 GPU, for 160 epochs at $128^2$. The physics-loss variants are slower because the structure-function and flux losses add per-step FFT work; `fno` and `wrsd_fno` carry the FNO block's parameters.

| Variant | Parameters (M) | Mean train time (s) | Mean peak memory (MB) |
|---|---|---|---|
| vanilla | 2.321 | 6647 | 2960 |
| se | 2.371 | 6890 | 2985 |
| fno | 7.077 | 8161 | 3178 |
| fno_operator | 4.970 | 8387 | 2132 |
| wrsd_gate | 2.511 | 8738 | 3069 |
| wrsd_phys | 2.321 | 8789 | 2960 |
| **wrsd** | 2.511 | 11329 | 3087 |
| **wrsd_fno** | 7.266 | 12628 | 3243 |

**Inference cost.** Table 12 reports sampling throughput and peak inference memory, and compares them to the DNS solver. We time the generation of a batch of 64 fields at the 50-step default on one V100

(mean of three runs after a warmup), and the solver time to advance one decorrelation interval (200 RK4 steps), which produces one statistically independent field along a trajectory. A U-Net variant draws a field in 66–71 ms; the gate adds 3.5% over the plain U-Net (65.9 → 68.2 ms) for its 8% extra parameters, and the FNO branch a further 2.3 ms. The standalone FNO operator is the fastest at 35.9 ms but is the weakest on the physics metrics (Table 1). The DNS solver needs 1.41 s per decorrelated field, so the generator is about 20× faster per independent draw at this resolution, before counting the spinup a fresh DNS trajectory requires. As a second-hardware point, on CPU (no GPU) the same 50-step sampling takes 2.35 s/field for WRSG (2.24 for vanilla, 1.97 for the FNO operator), about 2.7× faster than the CPU DNS solver's 6.27 s per decorrelated field, so the per-independent-field advantage holds without an accelerator, at about 34× lower absolute throughput than the V100. Peak inference memory on the V100 is modest ($\approx 1.83$ GB for the U-Net variants, 1.68 GB for the FNO operator) and is dominated by the 50-step solver's activation history rather than the parameters; we did not instrument host-memory use for the CPU run.

Table 12: Inference cost on one V100: time to sample one $128^2$ field at 50 solver steps (batch of 64, mean of three runs), throughput, and peak inference memory. The DNS row is the solver wall-clock to advance one decorrelation interval (200 RK4 steps), i.e. one independent field along a trajectory.

| Variant | ms / field | fields / s | Peak mem (MB) |
|---|---|---|---|
| vanilla | 65.9 | 15.2 | 1832 |
| se | 66.4 | 15.1 | 1832 |
| fno | 68.3 | 14.6 | 1851 |
| fno_operator | 35.9 | 27.9 | 1681 |
| wrsd_gate | 68.2 | 14.7 | 1834 |
| wrsd_phys | 66.1 | 15.1 | 1832 |
| **wrsd** | 68.3 | 14.6 | 1833 |
| **wrsd_fno** | 70.6 | 14.2 | 1852 |
| DNS solver | 1409.8 | 0.7 | – |

## C  Downstream Use: Reconstruction from Sparse Observations

To test whether the spectral fidelity of the prior carries to a downstream task, we use each trained model as the prior in a sparse-reconstruction (data-assimilation) problem: given a random fraction of the pixels of a held-out DNS field, with small observation noise added, we reconstruct the full field by pinning the observed pixels to the noised observation at every reverse step and letting the prior fill the rest (Lugmayr et al., 2022). For the point estimate we report the posterior mean over an 8-member ensemble, which minimizes the expected pixel error. Table 13 reports the reconstruction error on the unobserved pixels and the log-spectral distance of the reconstruction, at three observation densities. Both diffusion priors reconstruct far better than a mean-fill baseline (RMSE 0.10–0.76 against 1.01 for the trivial fill), so the generative prior carries real information. Between the two priors, WRSG gives the lower error at every density on both metrics: lower RMSE (0.743 vs 0.761 at 10%, 0.104 vs 0.131 at 50%) and lower log-spectral distance (1.975 vs 2.025, 1.110 vs 1.221). A single posterior draw is noisier, and at the sparsest observation its per-pixel error is underdetermined, so the two priors are comparable on one draw; the posterior mean is the appropriate point estimate, and there the spectral advantage of the WRSG prior carries through to the pixel reconstruction at every density.

Table 13: Sparse-reconstruction downstream task at $\nu = 0.013$ (48 held-out fields, observation noise 0.02), posterior-mean reconstruction over an 8-member ensemble. RMSE is over the unobserved pixels; LSD is the reconstruction's log-spectral distance. Both diffusion priors beat the mean-fill baseline; the WRSG prior gives the lowest error at every density on both metrics. Lower is better; best (excluding the trivial mean-fill) in bold.

| Observed | Reconstruction RMSE | | | Reconstruction LSD | | |
|---|---|---|---|---|---|---|
| | mean-fill | vanilla | WRSG | mean-fill | vanilla | WRSG |
| 10% | 1.011 | 0.761 | **0.743** | 2.338 | 2.025 | **1.975** |
| 25% | 1.010 | 0.454 | **0.416** | 2.399 | 1.844 | **1.777** |
| 50% | 1.010 | 0.131 | **0.104** | 2.376 | 1.221 | **1.110** |

## D Hyperparameter Ablations

This appendix sweeps the gate and training hyperparameters one at a time, holding the rest at the main-model setting (bins 16, rank 2, loss-weight scale $1\times$, $\sigma_{\max} = 20$, 50 sampling steps). Unless noted, each cell is trained for the full 160 epochs on three seeds $(29, 47, 89)$ and evaluated on the fixed test set.

**Sampling steps.** Table 14 re-samples the trained checkpoints at solver step counts from 10 to 100. The aggregate LSD is flat above about 20 steps for every variant, so the 50-step default is not load-bearing.

Table 14: Aggregate LSD against the number of DPM-Solver++ steps, mean over five seeds, on a fixed 192-field test subset (hence the small offset from Table 1). The LSD changes by under 0.01 above 20 steps.

| Steps | 10 | 15 | 20 | 30 | 50 | 75 | 100 |
|---|---|---|---|---|---|---|---|
| vanilla | 0.740 | 0.736 | 0.734 | 0.733 | 0.733 | 0.733 | 0.733 |
| **wrsd** | 0.297 | 0.292 | 0.291 | 0.290 | 0.289 | 0.289 | 0.289 |
| **wrsd_fno** | 0.300 | 0.286 | 0.283 | 0.281 | 0.281 | 0.280 | 0.280 |

**Radial bins and MLP rank.** Table 15 sweeps the bin count $\{8, 16, 32\}$ and the rank $\{1, 2, 4\}$, three seeds each. The aggregate LSD is flat across all five settings (0.288–0.300), so the gate is insensitive to these choices and the 16-bin, rank-2 default is not load-bearing; the 32-bin setting is marginally better on the flux and integral-length metrics. The binning is therefore not finely hand-tuned within its useful range.

Table 15: Gate bin/rank sweep (mean over three seeds). LSD is flat across all settings; the 32-bin gate is best on flux and integral length. Lower is better.

| Bins | Rank | LSD | LSD std | Energy flux | Integral len. err. |
|---|---|---|---|---|---|
| 8 | 2 | 0.300 | 0.008 | 0.082 | 0.102 |
| 16 | 1 | 0.299 | 0.009 | 0.086 | 0.113 |
| 16 | 2 | 0.299 | 0.010 | 0.085 | 0.116 |
| 16 | 4 | **0.288** | 0.003 | 0.084 | 0.111 |
| 32 | 2 | 0.299 | 0.004 | **0.062** | **0.097** |

**Physics-loss weights.** Table 16 scales all five weights by $\{0.5, 1, 2\}\times$. The $1\times$ setting minimizes the LSD (0.299); halving the weights raises the LSD to 0.320 but lowers the integral-length error to 0.086, the best of the three, and doubling them worsens both (0.413 and 0.132). The integral-length regression of Section 6 tracks the loss weight: the small-scale and transport terms buy structure-function fidelity ($S_2$ from 0.092 at

0.5× to 0.044 at 1×) at the cost of the large scales, and a practitioner who needs the integral length can trade it back by lowering the weights.

Table 16: Physics-loss-weight sweep (mean over three seeds), all five weights scaled jointly. The 1× setting is best on LSD and $S_2$; halving the weights is best on integral length. Lower is better.

| Scale | LSD | LSD std | $S_2$ log-RMSE | Integral len. err. | Energy flux |
|---|---|---|---|---|---|
| 0.5× | 0.320 | 0.014 | 0.092 | **0.086** | 0.094 |
| 1× | **0.299** | 0.010 | **0.044** | 0.116 | 0.085 |
| 2× | 0.413 | 0.010 | 0.052 | 0.132 | **0.080** |

**Localizing the integral-length regression.** Three interventions trace the integral-length regression to the structure-function loss and map the trade-off (Table 17). Up-weighting the integral-length loss alone, $\lambda_L \times \{2, 4\}$ with the other weights held, leaves the integral length flat ($0.116 \rightarrow 0.114$): the single-scalar integral-length target is a weak constraint and emphasizing it does not help. Adding a dense low-wavenumber spectral loss, matching the radial energy spectrum over $k \leq 6$ in log-space, lowers the integral length only marginally ($0.116 \rightarrow 0.108$) while sharply raising the LSD ($0.299 \rightarrow 0.766$ at weight 0.3) and lowering inverse-cascade recovery ($46.6\% \rightarrow 29.7\%$). Removing the structure-function loss entirely, by contrast, recovers the integral length completely, to 0.005, below even the plain U-Net's 0.039, but raises the LSD to 0.537 and halves the cascade recovery, because that term is what buys the small-scale and transport fidelity. The structure-function loss is the source of the regression, and the integral length trades directly against the small-scale fidelity it provides: the gate alone is integral-length-optimal (0.013, Table 1), and a practitioner can choose a point on this frontier by scaling that one term, with the 0.5× joint downscaling above (0.086 at LSD 0.320) a moderate compromise.

Table 17: Targeted interventions on the integral-length regression (mean over three seeds). Up-weighting $\lambda_L$ does not move the integral length and a low-wavenumber spectral loss moves it marginally at a large LSD cost; removing the structure-function loss recovers it fully, below the plain U-Net, but sacrifices the LSD and cascade. The structure-function loss is the source of the trade-off. Lower is better.

| Intervention | Integral len. err. | LSD | Inv. cascade % |
|---|---|---|---|
| Baseline ($\lambda_L 1\times$, full losses) | 0.116 | **0.299** | **46.6** |
| $\lambda_L 2\times$ | 0.114 | 0.296 | — |
| $\lambda_L 4\times$ | 0.115 | 0.309 | — |
| + low-$k$ loss 0.1 | 0.109 | 0.485 | 42.8 |
| + low-$k$ loss 0.3 | 0.108 | 0.766 | 29.7 |
| Structure loss off | **0.005** | 0.537 | 28.3 |

**Scalar loss configuration.** Table 18 sweeps the physics-loss configuration for the passive-scalar transfer of Section 5.8. Scaling the vorticity-tuned weights down lowers the scalar LSD monotonically toward the gate-only value, and restricting the loss to its field-agnostic spectral term (SPECTRAL-ONLY) matches the gate-only transfer (0.512 against 0.529, stable across five seeds at 0.512±0.050) while removing the regression of the full set (0.866). Adding the low-wavenumber term (SPECTRAL+low-$k$) hurts the scalar as it does the vorticity field (0.702).

Table 18: Scalar loss-configuration sweep (mean over three seeds, except SPECTRAL-ONLY over five), ordered by LSD. SPECTRAL-ONLY keeps the field-agnostic spectral term and drops the vorticity-specific ones; FULL ×$s$ scales all weights by $s$. Lower is better.

| Configuration | Scalar LSD | Integral len. err. |
|---|---|---|
| SPECTRAL-ONLY | **0.512** | 0.193 |
| gate only (no loss) | 0.529 | **0.194** |
| FULL ×0.25 | 0.591 | 0.302 |
| SPECTRAL+low-$k$ | 0.702 | 0.247 |
| FULL ×0.5 | 0.737 | 0.336 |
| FULL ×1.0 | 0.866 | 0.363 |

**Maximum noise level.** Table 19 sweeps $\sigma_{\max} \in \{10, 20, 40\}$ with the rest of the schedule fixed. The chosen $\sigma_{\max} = 20$ minimizes the LSD and the structure-function error: lowering it to 10 or raising it to 40 raises the LSD from 0.299 to 0.484 and 0.438 ($\sigma_{\max} = 40$ has a lower energy-flux error but a much worse LSD). With the pilot finding that the EDM default $\sigma_{\max} = 80$ collapses to near-zero samples (Section 4.5), this fixes $\sigma_{\max} = 20$ as a calibrated rather than arbitrary choice; $\sigma_{\max}$ and the training-noise center $P_{\mathrm{mean}}$ are coupled, so the recentered $\log \sigma \sim \mathcal{N}(-1.0, 1.2^2)$ accompanies the reduced range.

Table 19: Maximum-noise-level sweep (mean over three seeds), rest of the schedule fixed. The paper setting $\sigma_{\max} = 20$ minimizes the LSD and the structure-function error; $\sigma_{\max} = 40$ is lower on energy flux but much worse on LSD. Lower is better.

| $\sigma_{\max}$ | LSD | LSD std | $S_2$ log-RMSE | Energy flux |
|---|---|---|---|---|
| 10 | 0.484 | 0.020 | 0.335 | 0.129 |
| 20 | **0.299** | 0.010 | **0.044** | 0.085 |
| 40 | 0.438 | 0.021 | 0.125 | **0.051** |

# E   Qualitative Samples

Figure 11 shows paired samples from the same initial noise across the eight variants, with a DNS reference. The U-Net variants produce realistic vorticity fields with coherent cores and filaments; the standalone FNO operator is visibly grainier and less coherent, in line with its weaker physics metrics. The zoom shows the trade-offs already in the metrics: the plain baselines are slightly smoother, the physics-loss variants produce sharper small-scale detail, and WRSG and WRSG+FNO look alike despite the parameter gap.

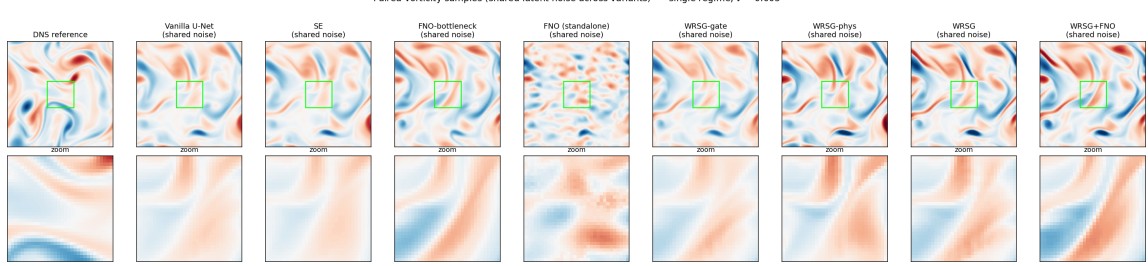

Figure 11: Paired vorticity samples from the same initial noise across variants, at $\nu = 0.005$. Top row: full $128 \times 128$ fields. Bottom row: the centered $32 \times 32$ patch. The DNS reference is one test draw, for orientation; the model panels are paired by latent noise.

## F  Training Dynamics

Figure 12 shows training and validation MSE. The seven U-Net variants converge to nearly the same MSE on the diffusion objective, so standard MSE validation does not separate architectures that produce visibly different samples. The standalone FNO operator sits above this cluster in both training loss and validation MSE, consistent with its being the weakest model on the physics metrics. Within the U-Net family the denoising MSE is flat while the spectral and cascade metrics in Table 1 vary widely, so we evaluate with the distribution-level metrics of Section 4.6.

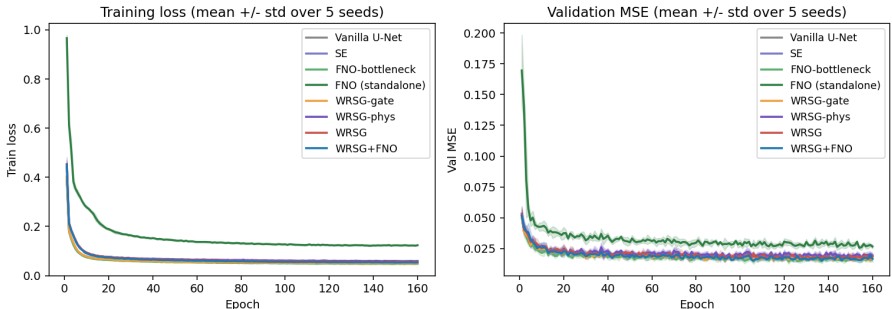

Figure 12: Training loss (left) and inner-validation MSE (right), mean ± standard deviation over five seeds. The seven U-Net variants converge to nearly the same denoising MSE despite their different sample statistics; the standalone FNO operator sits above the cluster.

## G  Structure Functions

Figure 13 shows the order-2 and order-3 vorticity structure functions against DNS; the ordering matches the log-RMSE values in Table 1.

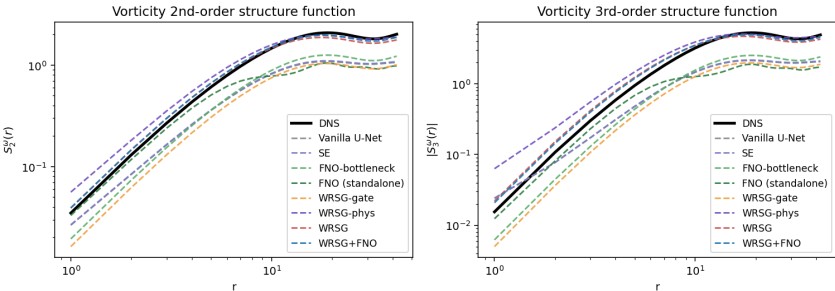

Figure 13: Vorticity structure functions, averaged over the test set. The physics-loss variants (WRSG-phys, WRSG, WRSG+FNO) match DNS most closely; the gate without the losses (WRSG-gate) sits furthest off.

## H  Transfer-Flow DNS Validation

The two transfer experiments of Section 5.8 use further direct numerical simulations, validated as for the main flow (Figure 1). The cellular-forcing DNS (Figure 14) shows the forcing peak at $k_f = 6$ and a resolved inertial range with a clean dissipation roll-off at $N = 128$, with the inertial slopes steepening with viscosity exactly as the Kolmogorov flow does. The passive-scalar DNS (Figure 15) shows the scalar variance developing through the spinup transient to a finite, statistically fluctuating level that is lower at higher diffusivity, and an inertial-convective scalar spectrum that steepens with diffusivity. Both confirm the transfer datasets are well-resolved DNS rather than under-resolved approximations.

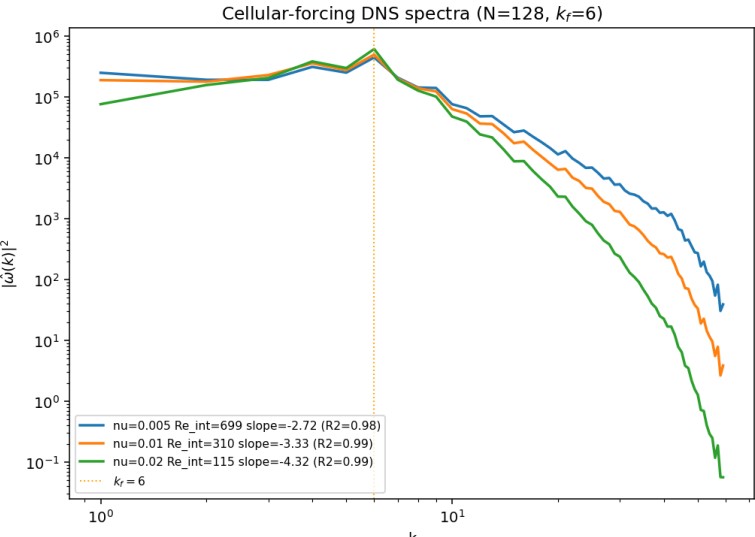

Figure 14: Cellular-forcing DNS ($N = 128$, $k_f = 6$): time-averaged vorticity power spectra at the three viscosity regimes ($\nu \in \{0.005, 0.010, 0.020\}$, $\text{Re}_{\text{int}} = 699/310/115$). The forcing peak is at $k_f = 6$ (dotted line) and the inertial-range slopes steepen from $-2.72$ to $-4.32$ with viscosity ($R^2 \geq 0.98$), as for the Kolmogorov flow (Figure 1).

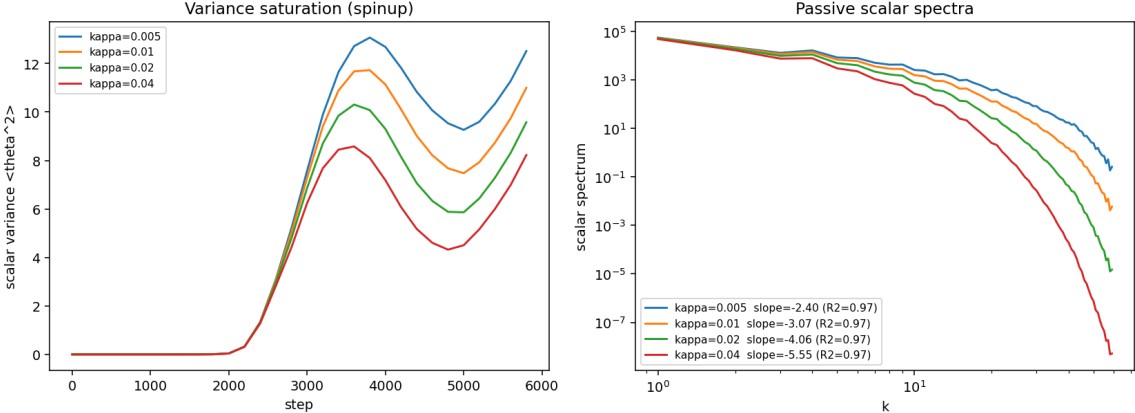

Figure 15: Passive-scalar DNS ($\nu = 0.010$): scalar variance $\langle \theta^2 \rangle$ through the spinup transient (left) and the time-averaged scalar power spectra (right) at the four scalar-diffusivity regimes ($\kappa \in \{0.005, 0.010, 0.020, 0.040\}$). The variance develops to a finite, statistically fluctuating level that decreases with $\kappa$; the scalar spectra show an inertial-convective range with slopes from $-2.40$ to $-5.55$ ($R^2 = 0.97$) steepening with $\kappa$.

## I  Conditional Rollout Comparison

To test the claim in the Discussion that unconditional sampling sidesteps the drift of a free-running conditional rollout, we train a standard conditional next-step diffusion emulator at $\nu = 0.013$ and compare its stationary distribution against ours. The emulator is a vanilla U-Net EDM denoiser (2.32M parameters, matching our vanilla baseline) that predicts the state 25 DNS steps ahead conditioned on the previous state, trained for 160 epochs on short-interval trajectory pairs with the same optimizer, schedule, and EMA as the

main models. We then roll it out autoregressively for 96 steps from real DNS initial conditions, discard a 32-step burn-in, and collect 512 stationary samples.

The emulator learns the one-step map well (validation MSE $6 \times 10^{-4}$), but its free-running distribution drifts: the vorticity spectrum accumulates excess energy in the dissipation tail (Figure 16), giving an aggregate log-spectral distance of 1.39 against DNS, against 0.49 for our unconditional WRSG samples at the same regime. The conditional rollout is also worse on the fluxes (energy 0.094 against 0.078, enstrophy 0.115 against 0.076) and recovers less of both cascades (inverse 37% against 51%, forward 44% against 86%). The drift is stable: rolled out to depth 200, the log-spectral distance holds at 1.3–1.4 across successive windows (steps 32–64, 80–128, 152–200) rather than growing or shrinking, so it is a stationary property of the free-running rollout rather than a burn-in transient, and an independently trained second seed reproduces it (LSD 1.43). This is the spectral drift that stabilizing methods such as Sambamurthy & Chattopadhyay (2025) are built to counter; the unconditional formulation avoids it by drawing independent stationary samples rather than integrating a trajectory. We use a standard emulator without rollout stabilization, so this compares a free-running conditional rollout against unconditional sampling, not the best possible conditional method.

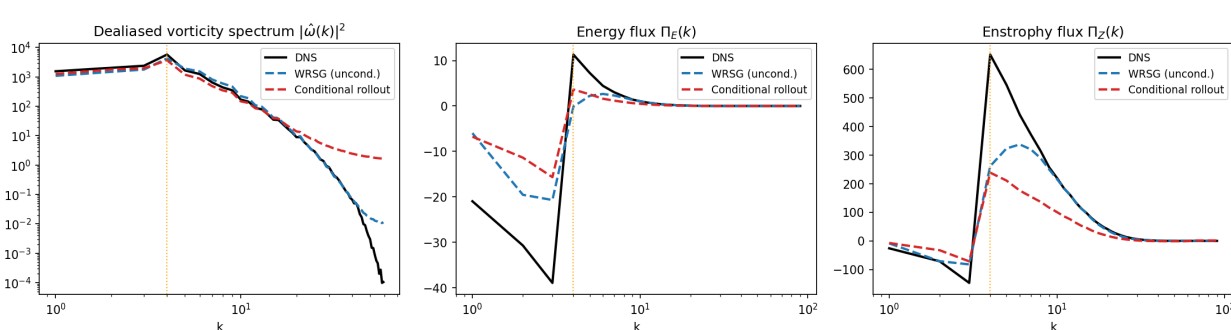

Figure 16: Unconditional WRSG against a standard conditional next-step emulator rolled out to a stationary state at $\nu = 0.013$ (512 samples each). Left: dealiased vorticity spectrum; the conditional rollout tracks DNS in the inertial range but accumulates excess energy in the dissipation tail. Center, right: energy and enstrophy spectral fluxes. The unconditional samples (aggregate LSD 0.49) stay closer to DNS across the board than the free-running conditional rollout (LSD 1.39).

## J   Short-Time Navier–Stokes Relaxation

The residual cascade gap (Section 6) is a failure of dynamical consistency: a snapshot-trained generator has no mechanism to enforce the nonlinear transport that sets the flux. We test whether a short pass of the true dynamics repairs it. Each generated field is mapped back to physical vorticity with the dataset normalization, advanced $\tau$ steps of the same pseudo-spectral solver that produced the data (its own viscosity per regime, the known forcing, RK4 at the training timestep), and re-normalized. This is a hybrid model–solver post-processor, not pure generation; it reads only the known viscosity and forcing, not the test fields, so it is also usable out of distribution. It leaves the headline deterministic samples untouched and defines a separate WRSG+NS variant. We report the relaxation trajectory over $\tau$ on the held-out test split as a characterization rather than selecting a single operating point (Figure 17); relaxing true DNS fields the same way leaves them at $\approx 100\%$ recovery with a log-spectral distance of 0.02, confirming the solver reuse is faithful.

The relaxation genuinely corrects the transport: WRSG+FNO's forward-enstrophy overshoot (here 107% on this experiment's test split, consistent with the 110% headline) falls toward the target, crossing 100% near $\tau \approx 210$ and reaching 93% at $\tau = 320$, while WRSG's inverse-cascade recovery climbs from 48% to 67%; the log-spectral distance also improves (WRSG+FNO $0.29 \to 0.27$, WRSG $0.30 \to 0.27$). It is a Pareto trade-off rather than a free repair: the same integration reshapes the small-scale phase structure, so the third-order

structure function $S_3$ degrades by 13–27% over the same $\tau$ range (Figure 17, right). No $\tau$ improves the cascade appreciably without a measurable $S_3$ cost, and the meaningful corrections need $\tau$ of order one eddy turnover of integration. A vanilla-U-Net prior relaxed the same way stays far worse at every $\tau$ (inverse 10% $\rightarrow$ 41%, forward 20% $\rightarrow$ 37% at $\tau = 320$, still below WRSG at $\tau = 0$), so the gain is not the solver alone: the generative prior supplies a near-equilibrium state that a short solver pass can polish, whereas a poor prior cannot be rescued in comparable time. The residual cascade error is thus a matter of dynamical consistency that a hybrid model–solver resolves at a structure-function cost, a concrete direction beyond the reach of a soft training penalty.

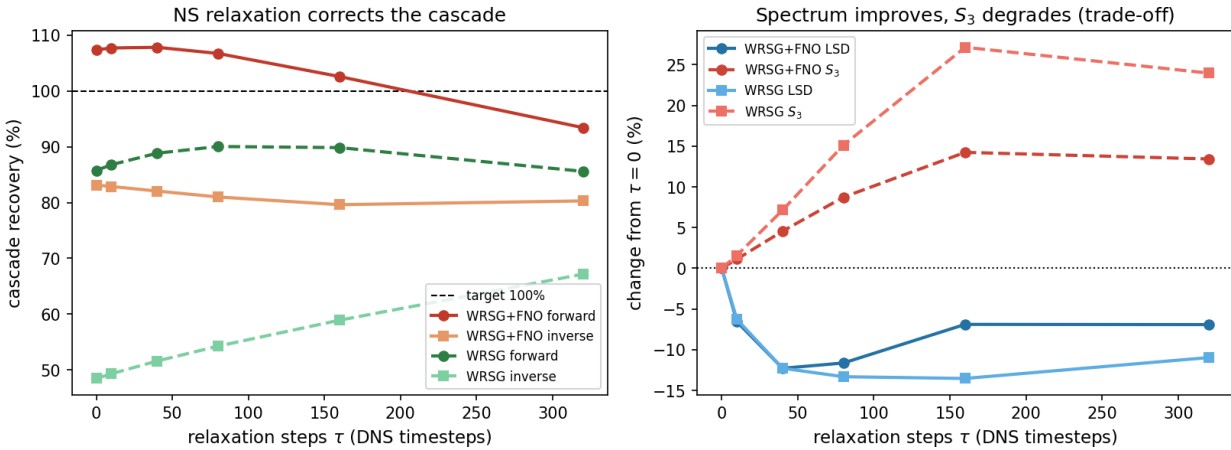

Figure 17: Short-time Navier–Stokes relaxation of generated fields (five-seed means; three seeds for the vanilla control; disjoint test split). Left: cascade recovery against relaxation length $\tau$; WRSG+FNO's forward overshoot corrects through 100% and both inverse cascades climb. Right: change from $\tau = 0$ in the log-spectral distance and the third-order structure function; the spectrum improves but $S_3$ degrades, a Pareto trade-off rather than a free repair.

