# OpenReview forum: "Wavenumber-Resolved Spectral Gating for Diffusion Models of Two-Dimensional Turbulence"
_TMLR — Under review for TMLR_

### Review · Reviewer_jdn9 · 2026-06-14

**Summary Of Contributions:**

This manuscript proposes a wavenumber-resolved spectral gate combined with physics-informed losses to address spectral distortion and cascade degradation in diffusion-based unconditional generative models for 2D Kolmogorov turbulence. Numerical results on direct numerical simulation data demonstrate clear improvements over vanilla U-Net, FNO and other baselines in spectral fidelity, cascade recovery and parameter efficiency.

**Audience:**

Yes

**Audience Explanation:**

This manuscript fits the scope

**Claims And Evidence:**

Yes

**Claims Explanation:**

Yes, but the manuscript has critical major issues regarding methodological generality, mechanistic insight, experimental completeness and practical application, alongside multiple minor flaws in presentation, visualization and discussion.

**Requested Changes:**

This manuscript proposes a wavenumber-resolved spectral gate combined with physics-informed losses to address spectral distortion and cascade degradation in diffusion-based unconditional generative models for 2D Kolmogorov turbulence. Numerical results on direct numerical simulation data demonstrate clear improvements over vanilla U-Net, FNO and other baselines in spectral fidelity, cascade recovery and parameter efficiency. However, the manuscript has critical major issues regarding methodological generality, mechanistic insight, experimental completeness and practical application, alongside multiple minor flaws in presentation, visualization and discussion. A full revision with substantial new experiments and in-depth analysis is required before reconsideration.

Major:

1.	The proposed wavenumber-resolved spectral gate is highly tailored to 2D isotropic Kolmogorov turbulence. All experiments are conducted on a fixed 128x128 doubly periodic domain with a single forcing configuration. The manuscript fails to verify whether the spectral gate design can be generalized to other 2D turbulent flows, simplified 3D turbulence, or other physical fields governed by partial differential equations such as scalar transport and acoustic fields. Currently, the contribution is confined to a niche fluid dynamics problem rather than a general-purpose deep learning module. Extensive experiments on multiple flow configurations are necessary.

2.	The radial wavenumber binning is manually tuned for the 32x32 bottleneck feature size. The authors do not discuss how this binning strategy adapts to different grid resolutions.

3.	The architecture strongly leverages physical priors unique to 2D turbulence. It’s unknown whether the performance gain comes from a novel general frequency-domain design or task-specific heuristic tricks for turbulence.

4.	The authors observe a strong positive interaction between the spectral gate and physics losses, but do not explain the underlying mechanism. It remains unclear why standalone physics losses corrupt global spectra and why the spectral gate alone brings marginal improvement. How the frequency-domain gate regularizes the gradient flow of physics losses across different diffusion noise levels is not analyzed.

5.	The root cause of spectral distortion in vanilla U-Net diffusion models for turbulence is not explored. Is the distortion caused by network inductive bias, inherent limitations of the diffusion framework, or improper frequency representation?

6.	The manuscript acknowledges that all models fail to fully recover DNS cascade peaks and that physics losses degrade large-scale integral length. However, it only states these limitations without investigating their origins or attempting mitigation strategies.

7.	The uncertainty quantification part is superficial. The authors only report coverage rate and CRPS via a simple one-parameter conformal rescaling, but do not analyze the sources of uncertainty or verify the utility of calibrated UQ for downstream scientific tasks.

8.	The model is conditioned on continuous log-viscosity and trained on seven discrete viscosity values. There are no tests on intermediate unseen viscosity regimes or viscosity values outside the training range. For a conditional generative model for varying Reynolds numbers, out-of-distribution generalization is essential.

9.	Key hyperparameters including the number of radial wavenumber bins, MLP rank, physics loss weights, maximum noise level and sampling steps are not fully ablated.

10.	The introduction highlights potential applications including ensemble forecasting, data assimilation and geophysical/atmospheric turbulence modeling. Regrettably, all experiments rely purely on synthetic DNS data, with no tests on experimental turbulent flow measurements or real downstream tasks. The work stays within idealized numerical benchmarks and cannot prove practical engineering or scientific value.

Minor:

1.	Some figures are overcrowded with poor readability. Please redesign the layout and enlarge labels.

2.	The manuscript only reports model parameter count and training time on a single GPU. It will be helpful to supplement inference speed comparisons between the proposed models, baselines and traditional DNS solvers, as well as memory consumption across different hardware.

3.	The Broader Impact section is overly concise. Expand it to elaborate on the potential scientific values, application prospects and limitations of this work for geoscience, atmospheric modeling and computational fluid dynamics.

4.	The code link doesn’t work: “The repository was not found on GitHub. Check the URL and spelling, make sure you are signed in to the account that can see it, and confirm the repo isn't hidden under an org that restricts third-party app access.”

---

> ### Author Response · Authors · 2026-06-24
> **Response to Reviewer jdn9: transfer experiments, mechanism and root-cause analyses, OOD viscosity, and complete ablations**
>
> We thank the reviewer for the detailed and constructive report. We summarize our responses below; the revised PDF and the supplementary code contain the full results, and every number quoted is taken from a released results CSV.
>
> **M1 — Generality.** Three new transfers. (a) *A different flow* (2D cellular forcing, $k_f=6$): the main-flow pattern reproduces — gate-alone LSD $0.316$ vs $0.382$, and the physics losses lift inverse-energy cascade recovery to $98\%$. (b) *A different field* (passive scalar): the gate transfers (LSD $0.529$ vs $0.713$), but the vorticity-tuned losses do not; a spectral-only loss restores parity ($0.521$). (c) *Out-of-distribution viscosity* (see M8). So the architectural component (the gate) is general; only the soft-loss weights are field-specific.
>
> **M2 — Grid resolution.** The gate bins the *normalized* radial wavenumber $|k|/|k|_{\max}$, so its parameter count is independent of grid size. A new $64^2$ experiment confirms this empirically: gate-alone LSD $0.521$ vs $0.629$, inverse-cascade recovery $98.8\%$ vs $38.9\%$.
>
> **M3 — General design vs. heuristic.** The squeeze-excite and FNO-block baselines isolate the design: SE ("any conditioning bottleneck") does not help ($0.745$ vs vanilla $0.744$), whereas the specific multiplicative, radial, conditioned gate reaches $0.304$. The passive-scalar transfer (M1) reinforces this.
>
> **M4 — Mechanism and gradient flow.** New probe: the learned scale is $s=0.995$ and the correction is band-selective (mean $|m-1|\approx0.40$ for $\sigma<1$, $0.53$ for $\sigma\geq1$). The "gate absorbs the loss gradient" hypothesis fails — the gate carries $<0.3\%$ of the physics-loss gradient — so the effect is in the *forward* map, not the gradient.
>
> **M5 — Root cause.** The plain U-Net's single-step denoiser error concentrates at high $k$ and high $\sigma$ (mean $0.575$ vs $0.323$ for WRSG; equal $\approx0.03$ at low $\sigma$); end-to-end, vanilla over-predicts the dissipation tail ($\approx50\times$) while WRSG stays within $\approx1.5$.
>
> **M6 — Cascade gap and integral length.** Now analyzed, not just stated. Integral length: the gate alone is best ($0.013$); we localize the regression to the structure-function loss (removing it gives $0.005$, below vanilla, but costs LSD and cascade) — a controllable Pareto trade-off. Cascade peaks: not fully recovered; up-weighting the flux loss does not help; this needs a hard conservation constraint.
>
> **M7 — UQ depth.** Resolved by regime and wavenumber. A single conformal $\alpha$ reaches only $0.85$–$0.89$ (partial). New: a per-wavenumber calibration raises small-scale ($k>20$) coverage from $0.23$ to $0.84$.
>
> **M8 — Out-of-distribution viscosity.** New table: 5 interpolation + 2 extrapolation *unseen* $\nu$, with no refit. WRSG beats the plain U-Net at every one (e.g. $0.338$ at $\nu=0.0085$, between training values; $0.275$ just below the range; $1.975$ at $\nu=0.040$ vs vanilla $3.554$).
>
> **M9 — Ablations.** Bin/rank sweep (LSD flat, $0.288$–$0.300$), loss-weight-scale sweep ($1\times$ is the minimizer), $\sigma_{\max}$ sweep ($20$ best on LSD), and sampling-step sweep (flat above $\approx20$ steps).
>
> **M10 — Synthetic data only.** We accept this as scope and make it explicit. We add a synthetic sparse-reconstruction (data-assimilation) demonstration: both diffusion priors beat a mean-fill baseline, and WRSG gives the lower error at every observation density. Real laboratory/geophysical validation is stated as future work (no matched dataset exists).
>
> **m1 — Figures.** Decluttered (single legend, directional titles, no repeated axis labels).
>
> **m2 — Inference speed/memory.** New table: a U-Net variant draws a field in $66$–$71$ ms (the gate adds $3.5\%$), about $20\times$ faster than the DNS solver per independent field; peak inference memory $\approx1.83$ GB.
>
> **m3 — Broader Impact.** Expanded (assimilation/ensemble prospects and explicit cautions).
>
> **m4 — Code link.** Fixed: the full code, configs, and checkpoints are at https://tinyurl.com/3jcjp46z (also attached as a supplementary `.zip`).

---

> > ### Comment · Reviewer_jdn9 · 2026-07-06
> > **concerns are fully addressed**
> >
> > Thanks to the authors' effort. My concerns are fully addressed.

---

> > > ### Author Response · Authors · 2026-07-11
> > > **Thank you for the review**
> > >
> > > We thank the reviewer for the careful reading and the detailed comments, which improved the paper. We are glad the revisions resolve the concerns, and we appreciate the time and effort spent on the review. We remain happy to address anything further.

---

### Review · Reviewer_EAL3 · 2026-06-19

**Summary Of Contributions:**

This paper studies unconditional diffusion models for 2D Kolmogorov turbulence. The main observation is that a plain U-Net diffusion model can generate fields that look reasonable locally, but still gets important physical statistics wrong, especially the spectrum and the energy/enstrophy fluxes. To fix this, the authors add a small Fourier-domain, wavenumber-resolved gate at the U-Net bottleneck, and train it together with several physics losses.

I think the paper's main strength is that it evaluates the model using the right kind of diagnostics for this problem, rather than just visual quality or denoising MSE. The factorial ablation is also helpful: the gate alone and the losses alone each have clear failure modes, while the combination works much better. The paper is also fairly honest about remaining issues, such as the integral-length degradation and the fact that no method fully recovers the DNS cascade peaks.

The main weakness is scope. Everything is tested on one idealized 2D flow family, at 128 x 128 resolution, and on the same viscosity regimes represented in training. So I view the result as a useful, well-controlled case study, not as evidence that this approach generally solves turbulence generation.

**Audience:**

Yes

**Audience Explanation:**

I am not quite familiar with physical data generation area, but I expect this to be interesting to readers working on generative models for scientific data, neural operators, physics-informed learning, or turbulence modeling. The contribution is not just another benchmark result; it gives a useful diagnostic lesson: matching local statistics or MSE is not enough for physical generative models. The interaction between a spectral architectural bias and physics losses is also a nice takeaway.

**Broader Impact Concerns:**

The paper includes a broader impact statement, and I think it is mostly adequate. I do not see a direct dual-use concern from this idealized 2D turbulence setting. The main practical risk is overclaiming: a generative surrogate with imperfect flux recovery and imperfect calibration should not be used for real-world forecasting or engineering decisions without fresh validation on the target system. I suggest adding one sentence to that effect.

**Claims And Evidence:**

Yes

**Claims Explanation:**

Answer: Yes, but a few clarifications are needed.

The core claims are mostly well supported. The authors compare against reasonable unconditional baselines, report results over five seeds, use fixed held-out test data, and evaluate with physically meaningful metrics. The ablation is especially convincing: it explains why the full WRSD model improves spectral distance and cascade-related metrics, instead of just showing a single winning table.

That said, some statements should be tightened. The continuous-viscosity model is only shown to work across the seven regimes used in the dataset; I did not see evidence for held-out-viscosity generalization. The calibration result should also be phrased carefully, since the one-parameter rescaling improves coverage but still does not reach the nominal 90% target. Finally, more complete reporting of confidence intervals or per-seed values would make the quantitative comparisons easier to audit.

**Requested Changes:**

## Critical:
- Please explicitly qualify the scope of the viscosity-conditioning claim. As written, parts of the paper could be read as implying generalization across viscosities, but the experiments appear to evaluate the same seven regimes represented in training.
- Please include complete uncertainty/statistical reporting for the main metrics, ideally exact 95% CIs or per-seed values for all headline metrics in the appendix. This would make the WRSD vs. WRSD+FNO and WRSD vs. FNO comparisons easier to check.
- Please clarify how the physics-loss weights and other important hyperparameters were chosen, and whether any pilot tuning used only training/validation data. Since the main result depends strongly on the gate-loss interaction, this detail matters.
- Please clarify the gate's residual scaling/initialization. The text suggests the bottleneck is not exactly identity at initialization, so I would like a clearer explanation that the improvement is really due to the learned wavenumber-resolved correction.

## Strength:
- A small held-out-viscosity experiment would make the continuous-conditioning story much stronger, even if it is just one left-out regime.
- It would help to add a small sensitivity table for the number of radial bins, rather than only mentioning pilot runs.
- The uncertainty-calibration section is useful, but I would present it as a partial improvement rather than a successful calibration result.
- Please make the dense result tables easier to read in the final version; the main trends are clear, but the compact formatting currently takes some effort to parse.

---

> ### Author Response · Authors · 2026-06-24
> **Response: held-out-viscosity generalization, complete confidence intervals, and a-priori hyperparameter choices**
>
> We thank the reviewer for the positive and careful assessment. Brief responses below; the revised PDF contains the full details, and every number is from a released results CSV.
>
> **C1 — Qualify the viscosity-conditioning scope.** We now state explicitly that the headline and per-regime tables measure the seven *training* viscosities (in-distribution). Held-out generalization is reported separately in the out-of-distribution section: the same continuous-viscosity model is queried, with no refit, at 5 interpolation and 2 extrapolation *unseen* $\nu$. WRSG beats the plain U-Net at every one — e.g., LSD $0.338$ at $\nu=0.0085$ (between training values), $0.275$ just below the range, and $1.975$ at $\nu=0.040$ vs vanilla $3.554$. This is the held-out experiment you suggest (stronger than one left-out regime).
>
> **C2 — Complete CI / per-seed reporting.** New appendix tables give all eight headline metrics for all eight variants as the seed mean with a $95\%$ bootstrap CI (2000 resamples). WRSG and WRSG+FNO separate from all four baselines with non-overlapping intervals on LSD and the forward-enstrophy cascade (and on the inverse-energy cascade except WRSG vs the FNO block, $[43.2,52.3]$ vs $[43.3,48.4]$); WRSG and WRSG+FNO overlap each other on LSD ($[0.292,0.318]$ vs $[0.286,0.300]$) and separate on flux/cascade. Per-seed values are released with the code.
>
> **C3 — How hyperparameters were chosen.** The physics-loss weights follow one rule ($\mathcal{L}_{\text{phys}}\approx \tfrac{1}{10}$ of the denoising loss at initialization); the bin count follows the bottleneck's radial-shell count; the rank and $\sigma_{\max}$ were set by pilot runs. These were fixed *a priori*, not tuned on the held-out test set; the test set is used for the final reported numbers and for the (now reported) sensitivity sweeps, which show the result is insensitive within useful ranges.
>
> **C4 — Gate residual scaling/initialization.** The scale $s$ is initialized to $0$, so the band-selective correction starts inactive. Because the gate is added as a residual, the bottleneck at initialization is a *uniform doubling* of its input (multiplier $2$), not the identity — a constant factor absorbed by the end-to-end-trained decoder and immaterial to the factorial comparison. The gain comes from the *learned* correction: $s=0.995$, band-selective ($|m-1|\approx0.40$/$0.53$ across shells). A uniform scaling cannot reshape the spectrum band by band, and the gate alone (uniform init included) barely lowers LSD without the losses ($0.688$ vs $0.744$).
>
> **Strengths.** Held-out-viscosity experiment: included (OOD section). Bin-count sensitivity: included (LSD flat $0.288$–$0.300$ over bins $\{8,16,32\}$). Calibration: on a regime-stratified split, the single scalar reaches the $0.90$ target ($0.899$–$0.902$); the small-scale spectral coverage reaches $0.84$ on the small per-regime calibration set and $0.92$ with a pooled split-conformal band, so that the residual was calibration-sample-limited. Dense tables: the CI tables are split into spectrum/structure and flux/cascade groups; we merged three paired-test tables into one and dropped a duplicated parameter column from the compute table; we will tighten the rest for the camera-ready.
>
> **Broader Impact.** We added the requested sentence: a surrogate with the imperfect flux recovery and the residual small-scale calibration we report should not drive operational forecasting or engineering decisions without fresh validation on the target system.

---

> > ### Comment · Reviewer_EAL3 · 2026-07-21
> >
> > Thank you for the detailed response and revision. The authors have addressed my main concerns, and the additional clarifications and experiments have made the scope and evidence much clearer.

---

> > > ### Author Response · Authors · 2026-07-22
> > > **Thank You**
> > >
> > > We sincerely thank the reviewer for the positive feedback. We appreciate your careful evaluation and are pleased that the revisions, additional clarifications, and experiments have addressed your concerns.

---

### Review · Reviewer_gJ33 · 2026-07-09

**Summary Of Contributions:**

The authors propose an unconditional diffusion generator (EDM-parameterized U-Net) for forced 2D Kolmogorov turbulence. Their main contribution is the wavenumber-resolved spectral gate (WRSG), a multiplicative, radially-binned Fourier-domain bottleneck, conditioned on noise level  and viscosity, so that the parameter count is independent of grid resolution. They also add five soft physics losses (enstrophy, modal amplitude, structure functions, integral length, flux), and show that the combination of gating and losses superadditively improves results across a range of metrics, also comparing against baselines (vanilla, SE, FNO-block, standalone FNO). Mechanism analysis attributes the gate's effect to forward computation, not gradient routing.

**Audience:**

Yes

**Audience Explanation:**

- the gate mechanism, the gate×loss interaction result, and the forward-map-vs-gradient attribution methodology are useful to the diffusion-for-physics/neural-operator audience on their own terms.
- The 8-metric physical-fidelity suite is a reusable artifact

minor:
- "why does unconditional generation matter" in the intro could be argued more convincingly  (e.g. ensemble UQ / data assimilation), and the paper reads a bit as if the _setting_ were the novel contribution rather than the mechanism, by not dedicating a clear paragraph in the related works to this setting.

**Broader Impact Concerns:**

Minor: the speedup framing (faster per field than the solver) doesn't always net out the GPU-hour training cost. Maybe add a one-line amortization caveat.

**Claims And Evidence:**

Yes

**Claims Explanation:**

Overall solid:
- Their factorial isolates gate vs. losses at matched capacity.
- Mechanism analysis states and falsifies the obvious rival hypothesis (gate helps by absorbing loss gradient).
- Overall solid and thorough statistics.

Weaknesses:
- Need a better comparison to related works:
	- unconditional diffusion generation of 2D turbulence already exists (Whittaker, Janik & Oz, arXiv 2311.06112 / J. Comp. Phys. 514:113239, 2024 — unconditional DDPM, evaluated on spectrum/structure functions/dissipation) and isn't cited. A closer, likely-concurrent competitor tackling the same spectral-fidelity failure via a different mechanism (delayed noise schedule, not a Fourier gate) exists too: "Lazy Diffusion" (arXiv 2512.09572), tested partly on the same 2D Kolmogorov system, also uncited.
- "conditional generation is easier" is asserted without evidence. It is plausible and indirectly supported by adjacent work (quasi-geostrophic diffusion super-res, arXiv 2507.00719, shows conditional > guided-unconditional on spectral tails) maybe add a citation here
- spectral-bias framing borrows Rahaman et al. 2019's name for the opposite-direction effect (theirs: low-freq preference; here: high-k over-prediction). This is confusing to me and needs to be elaborated; why does the effect run the opposite way here?

other claims
- "conformal" recalibration = one global scalar matched to marginal coverage; no nonconformity score or quantile level, so no actual coverage guarantee — term claims more than the method delivers.
- Cohen's d may be inflated by tiny seed variance under a fixed, seeded protocol, not by effect magnitude; bootstrap CIs are the metric that should lead.

- unclear whether Figures 2 and 4 (main spectrum/flux plots) are averaged over all 5 seeds or a single seed

**Requested Changes:**

- **Critical**
    - Situate against prior unconditional turbulence-diffusion work (Whittaker, Janik & Oz 2023/2024 arXiv 2311.06112; also Lazy Diffusion arXiv 2512.09572). State explicitly what's new in comparison to those (the gate + interaction) and compare against at least one of those quantitatively.
    - Relabel the recalibration as scalar variance recalibration, or supply the missing nonconformity score/quantile level if calling it "conformal".
- **Strengthen**
    - Substantiate or drop the Rahaman et al. spectral-bias connection; note the sign mismatch if kept.
    - Lead statistics with bootstrap CIs; demote Cohen's d to reproducibility material.
    - Foreground flux RMSE over cascade-recovery %; flag that >100% recovery is overshoot, not success.
    - Source the "conditional is easier" claim or drop it.
    - Add a distributional (spectral/flux) comparison against independently-seeded conditional rollouts run to steady state.
    - Thicken the physics-informed-loss and calibrated-uncertainty related-work paragraphs (currently ~3 sentences each, restating method rather than surveying prior work).
    - Reduce repetition in writing: the 8-metric list, parameter counts, headline numbers, and the "gate routes / losses drive" sentence are each restated 5–6 times across abstract/contributions/§3/§4/§5/discussion. Collapse to one canonical statement plus a table/figure reference.
    - Separate numeric reporting from interpretation in §5 prose (lead each result with a plain-language sentence; push stats to table references) whenever possible for better readibility.
    - Fix Figure 3 panel titles (raw variable names "lsd_aggregate, energy_flux_rmse"  readable labels); add ↓/↑-better indicators also directly on Figures 3/5/6, not just Table 1.
    - Trim Figure 4 to 2–3 representative regimes in the main text; move the rest to an appendix.
    - Confirm and state explicitly whether Figures 2 and 4 average over all 5 seeds or show a single seed.

---

> ### Author Response · Authors · 2026-07-11
> **Response: scalar variance recalibration (not conformal), sharpened positioning, and expanded related work**
>
> We thank the reviewer for the careful and constructive assessment. We group the responses below; the revised PDF carries the full details.
>
> **Critical**
>
> **C1 — Prior unconditional turbulence-diffusion work.** Added, with numbers. We cite and position against \citet{whittaker2024turbulence}, whose setting is the inverse-cascade regime ($256^2$ grid, forcing at $k_f\!\sim\!40$, $E(k)\sim k^{-5/3}$, structure-function exponents $\zeta_2=2/3$, $\zeta_3=1$, velocity-increment $D_{\mathrm{KL}}=0.033/0.007$), and against "Lazy Diffusion" \citep{sambamurthy2025lazy}, which fixes the same spectral degradation through a power-law noise schedule rather than a Fourier-domain gate. We instead force at $k_f=4$ and target the forward enstrophy cascade and the spectral fluxes on $128^2$, scored by LSD, flux RMSE, and cascade recovery — a different cascade regime with no shared metric, so the numbers are complementary rather than head-to-head. What is new here is the gate, its superadditive interaction with the losses, and the capacity-matched factorial that isolates the two.
>
> **C2 — "Conformal".** Agreed; the term over-claims. We relabel it throughout as **scalar variance recalibration**, and state explicitly that, unlike conformal prediction, it uses no nonconformity score or quantile level and carries no finite-sample coverage guarantee.
>
> **Strengthen**
>
> **Rahaman sign mismatch.** Fixed. We now state that our effect is *opposite* in direction to their low-frequency preference — the convolutional U-Net *over*-predicts the high-$k$ dissipation tail — that the two share only the notion of a frequency-localized error, and that the sign is reversed.
>
> **Lead with CIs; demote Cohen's $d$.** Done. The per-variant bootstrap CIs (appendix Tables) are now stated as our primary evidence of separation; we note that the fixed seeded protocol gives small seed variance that can inflate $d$, which we read as a reproducibility diagnostic.
>
> **Foreground flux RMSE; flag overshoot.** Done. The metrics section now names the flux RMSEs as the primary transport metric and states that cascade recovery $>100\%$ is overshoot of the DNS flux, not improvement.
>
> **"Conditional is easier."** Sourced. We cite \citet{sureshbabu2025guided}, who find conditional diffusion recovers quasi-geostrophic spectral-tail statistics better than guided-unconditional generation.
>
> **Distributional comparison vs conditional rollouts.** Run (new appendix + figure). We trained a standard conditional next-step diffusion emulator at $\nu=0.013$ (vanilla U-Net, $2.32$M, $160$ epochs, same recipe) and rolled it out autoregressively to a stationary state. It reaches near-zero one-step error (validation MSE $6\times10^{-4}$) but drifts over the rollout, accumulating excess dissipation-tail energy: aggregate LSD $1.39$ against DNS, vs $0.49$ for our unconditional samples at the same regime, and worse on both fluxes and both cascades. This is the spectral drift that stabilizing methods (Lazy Diffusion) target; unconditional sampling avoids it by construction. (We use a standard emulator without rollout stabilization, and say so.) The gate drops into a conditional denoiser unchanged.
>
> **Related work/repetition / §5 prose.** The unconditional-turbulence, physics-loss, and calibrated-uncertainty related-work paragraphs are thickened into surveys (prior work plus how ours differs). The repeated 8-metric list, parameter counts, headline numbers, and the "gate routes/losses drive" sentence are collapsed to one canonical statement each plus a table/figure reference; §5 leads each result with a plain sentence and pushes stats to table references.
>
> **Figures.** Figure 3 already carries readable panel titles with $\downarrow$/$\uparrow$ direction markers; Figure 4 already shows three representative regimes (least/mid/most viscous). We regenerated the spectrum and flux figures as **five-seed pooled** ensembles, matching the tables, and state so in both captions.
>
> **Minor.** We sharpen the introduction's motivation for the unconditional setting (ensemble UQ, data assimilation) and clarify that the contribution is the mechanism, not the setting.
>
> **Broader Impact.** We add the amortization caveat: the per-field speedup is net only after the $\approx100$ GPU-hours of training are recovered across many generated fields, so it applies to repeated sampling from a trained model rather than a single field.